# A DEMON AT WORK: LEVERAGING NEURON DEATH FOR EFFICIENT NEURAL NETWORK PRUNING

## ABSTRACT

When training deep neural networks, the phenomenon of 'dying neurons' —units that become inactive and output zero throughout training—has traditionally been viewed as undesirable, linked with optimization challenges, and contributing to plasticity loss in continual learning scenarios. In this paper, we reassess this phenomenon through the lens of network sparsity and pruning. By systematically exploring the influence of various hyperparameter configurations on the occurrence of dying neurons, we unveil their potential to facilitate simple yet effective structured pruning algorithms. We introduce 'Demon's Pruning' (DemP), a method that controls the proliferation of dead neurons, dynamically sparsifying neural networks as training progresses. Remarkably, our approach, characterized by its simplicity and broad applicability, outperforms existing structured pruning techniques, while achieving results comparable to prevalent unstructured pruning methods. These findings pave the way for leveraging dying neurons as a valuable resource for efficient model compression and optimization.

## 1 INTRODUCTION

Dying neurons, a phenomenon frequently observed during the learning process of neural networks, are traditionally viewed as detrimental, often leading to suboptimal performance (Maas et al., 2013; Xu et al., 2015) or loss of plasticity, especially in non-stationary settings (Lyle et al., 2023; Abbas et al., 2023). In response, alternative activation functions without a hard-saturated state, such as Leaky ReLU (Maas et al., 2013), Swish (Ramachandran et al., 2018), GELU (Hendrycks & Gimpel, 2016), have been proposed.

In this work, we reexamine the phenomenon of dying neurons through the lens of network sparsity and pruning. Building upon both intuitive and theoretical insights into neuron death within networks trained using stochastic optimization methods, we demonstrate how varying hyperparameters such as learning rate, batch size, and L2 regularization parameter influence the occurrence of dead neurons during training. We present and validate a method for actively managing the emergence of dead units and for dynamically pruning them throughout the training process.

Notably, we observe that a higher level of noise or stronger regularization leads to sparser solutions, characterized by a higher number of dead neurons. Capitalizing on the simplicity of our pruning criterion –removing the inactive neurons– we introduce at no additional cost a structured pruning method, *Demon Pruning* (DemP), both performant and easy to implement. DemP can be seamlessly integrated into any training algorithm and readily combined with existing pruning techniques.

DemP marks a significant departure from traditional methodologies in pruning. Previous methods relied on heuristics-based interventions: training is paused to ablate weights (or neurons) based on a specific criterion. Training then resumes and tries to recover from the intervention. In contrast, DemP is the first instantiation of a potential family of algorithms that leverage insights into how the interplay between stochasticity and sparsity affects learning dynamics. With DemP, the optimization process directly leads to sparse networks, removing the need for direct interventions during training. Moreover, because the neurons removed by DemP in ReLU networks were inactive, the learning dynamics are not impacted by the pruning procedure, removing the need for recovery.

Structured pruning methods, even in the absence of specialized sparse computation primitives (Elsen et al., 2020; Gale et al., 2020), can more effectively exploit the computational advantages of GPU

hardware (Wen et al., 2016) compared to unstructured methods. This becomes particularly crucial as deep learning models continue to grow; as considerations for environmental impacts become increasingly significant (Strubell et al., 2019; Lacoste et al., 2019; Henderson et al., 2020), developing methods with reduced energy footprint that can be adopted widely is becoming fundamental.

Our main contributions are:

1. **Analysis of Neuron Mortality**: We provide insights into the mechanisms underlying neuron death, highlighting the pivotal role of stochasticity, as well as the influence of varying hyperparameters such as learning rate, batch size, and regularization parameters (Section 3).

2. **A Structured Pruning Method.** Leveraging our insights, we introduce DemP, a novel pruning approach that both promotes the proliferation of dead neurons in a controlled way, and removes dead neurons in real time as they arise during training, offering substantial training speedups (Section 4).

3. **Empirical Evaluation.** Through extensive experiments on various benchmarks, we demonstrate that DemP, despite its simplicity and broad applicability, surpasses existing structured pruning methods in terms of accuracy-compression tradeoffs, while achieving comparable results to prevalent unstructured pruning methods (Section 5).

## 2 RELATED WORKS

**Dead Neurons and Capacity Loss.** It is widely recognized that neurons, especially in ReLU networks, can die during training (Agarap, 2018; Trottier et al., 2017; Lu et al., 2019). In particular, Evci (2018) noted the connection of the dying rate with the learning rate and derived a pruning technique from it.

More recently, dead neurons were studied in continual and reinforcement learning through the lens of *plasticity loss* (Berariu et al., 2021; Lyle et al., 2022), that progressively makes a model less capable of adapting to new tasks Kirkpatrick et al. (2016). The inability to adapt has also been observed in supervised learning (Ash & Adams, 2020).

In some scenarios, a cause of plasticity loss has been attributed to the accumulation of dead units (Sokar et al., 2023; Lyle et al., 2023; Abbas et al., 2023; Dohare et al., 2021). These works have shown that under rapid shifts in training distribution, neural network activations can collapse to a region where the gradient is 0. Although ReLU activation seems to amplify the phenomenon, it has also been observed for various activation functions that have a saturated regime (Dohare et al., 2021). Simple solutions such as resetting the dead units (Sokar et al., 2023) or concatenating ReLU activations (Abbas et al., 2023) have proven effective to mitigate the issue.

**Sparsity Induced by Stochasticity.** Work by Pesme et al. (2021); Vivien et al. (2022) studied the impact of Stochastic Gradient Descent's (SGD) noise on training, following empirical observations that SGD can be beneficial to generalization over Gradient Descent (GD) (Keskar et al., 2017; Masters & Luschi, 2018). The noise structure of SGD (Wojtowytsch, 2023; Pillaud-Vivien, 2022) plays a key role in their observations.

**Pruning.** Pruning is used to reduce the size and complexity of neural networks by removing redundant or less important elements, be they neurons or weights while maintaining their performance (LeCun et al., 1989). Recent advances such as those based on the Lottery Ticket Hypothesis (Frankle & Carbin, 2019) have demonstrated the existence of subnetworks trainable to comparable performance as their dense counterpart but with fewer parameters. Pruning techniques can broadly be categorized into two groups: structured pruning and unstructured pruning.

*Structured pruning* aims to remove entire structures within a network, such as channels, filters, or layers. It results in smaller and faster models that maintain compatibility with existing hardware accelerators and software libraries (Wen et al., 2016; Li et al., 2017). We highlight and benchmark against recent works that use criteria based on gradient flow to evaluate which nodes to prune (Verdenius et al., 2020; Wang et al., 2020; Rachwan et al., 2022). Other works employed either L0 or L1 regularization on gate parameters (or batch normalization scaling parameters) to enforce sparsity (Liu et al., 2017; Louizos et al., 2018; You et al., 2019), but we do not benchmark them as they are outperformed by Rachwan et al. (2022).

*Unstructured pruning*, on the other hand, focuses on removing individual weight from the network (LeCun et al., 1989; Han et al., 2016b). This approach often leads to higher compression rates but requires specialized hardware or software implementations for efficient execution due to the irregularity of the resulting sparse models (Han et al., 2016a). One notable method in unstructured pruning is magnitude-based pruning (Han et al., 2015), where weights with magnitudes below a certain threshold are removed. More recent approaches include dynamic sparse training methods such as RigL (Evci et al., 2020; Lasby et al., 2023) and SNFS (Dettmers & Zettlemoyer, 2019), which iteratively prune and regrow connections during training based on their importance.

*Regularization-based pruning* has been popular for both structured and unstructured pruning, with canonical papers employing L0 or L1 regularization to induce sparsity directly (Louizos et al., 2018; Liu et al., 2017; Ye et al., 2018) while L2 can help identify the connections to prune with the smallest weight criterion (Han et al., 2015). Because uniform regularization can quickly degrade performance (Wen et al., 2016; Lebedev & Lempitsky, 2016), Ding et al. (2018) and Wang et al. (2019) proposed to adapt the regularization for different parameter groups. Recently, Wang et al. (2021) showed that growing the L2 regularization can leverage Hessian information to identify the filters to prune in pre-trained networks.

## 3 NEURAL DEATH: AN ANALYSIS

In this section, we study the phenomenon of dead neurons accumulation during training in deep neural networks. Our aim is to provide theoretical insights into this phenomenon and investigate how various training heuristics and hyperparameters affect neuron mortality.

Given a deep neural network and a set of $n$ training data samples, we denote by $a_j^\ell \in \mathbb{R}^n$ the vector of activations of the $j$th neuron in layer $\ell$ for each training input. We adopt the following definition of a "dead neuron" throughout the paper:

**Definition:** The $j$-th neuron in layer $\ell$ is *inactive* if it consistently outputs zero on the entire training set, i.e $a_j^\ell = 0$.[1] A neuron that becomes and remains inactive during training is considered as *dead*.[2]

Many modern architectures use activations functions with a saturation region that includes 0 at its boundary. In this case, when a neuron becomes inactive during training, its incoming weights also receive zero or very small gradients, which makes it difficult for the neuron to recover. In this paper, we mostly work with the Rectified Linear Unit (ReLU) activation function, $\sigma(x) = \max(0, x)$. In this case, the activity of a neuron depends on the sign of the corresponding pre-activation feature.

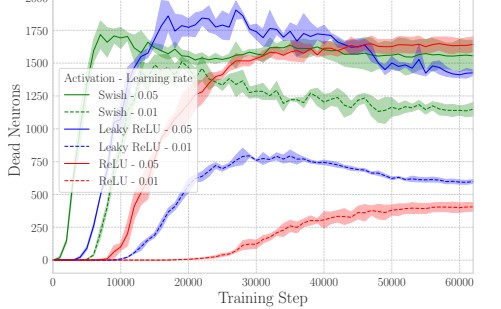

Figure 1: Dead neurons accumulation for a ResNet-18 trained on CIFAR-10.

The network with parameter $\boldsymbol{w}$ is trained to minimize the training loss $L(\boldsymbol{w}) = \frac{1}{n} \sum_{i=1}^{n} \ell_i(\boldsymbol{w})$, where $\ell_i(\boldsymbol{w})$ is the loss function on sample $i$, using stochastic gradient descent (SGD) based methods. At each iteration, this requires an estimate of the loss gradient $g(\boldsymbol{w}) := \nabla L(\boldsymbol{w})$, obtained by computing the mean gradient on a random minibatch $b \subset \{1 \cdots n\}$. For simple SGD with learning rate $\eta$, the update rule takes the form

$$\boldsymbol{w}_{t+1} = \boldsymbol{w}_t - \eta \hat{g}(\boldsymbol{w}_t, b_t), \quad \hat{g}(\boldsymbol{w}, b) := \frac{1}{|b|} \sum_{i \in b} \nabla \ell_i(\boldsymbol{w}). \tag{1}$$

### 3.1 NEURONS DIE DURING TRAINING

We begin with some empirical observations. Using the above definition with a fixed thresholding parameter ($\epsilon = 0.01$), we monitor the accumulation of dead neurons during training of a Resnet-18

---

[1]In practice, especially for non-ReLU activation functions, we will be using the notion of $\epsilon$-inactivity, defined by the condition $|a_i^\ell| < \epsilon$, for some threshold parameter $\epsilon$.

[2]For convolutional layers, we treat the filters as neurons, see Appendix A

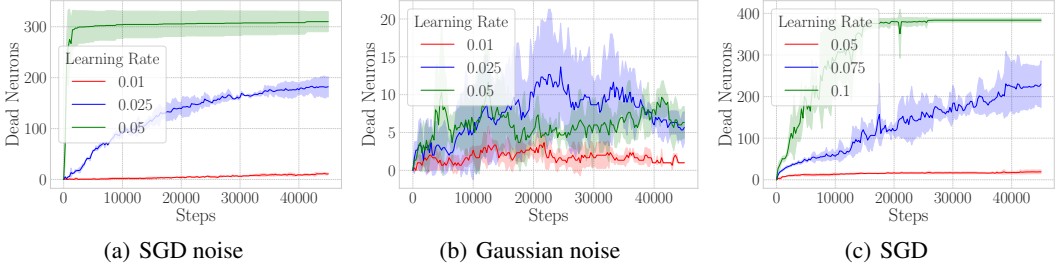

(a) SGD noise            (b) Gaussian noise            (c) SGD

Figure 2: A 3-layer MLP trained over a subset on MNIST. (a) The noisy part of the minibatch gradient is isolated and used exclusively to update the NN. It shows that noisy updates are *sufficient* to kill a subset of neurons following standard initialization. Because SGD gradient is 0 for dead neurons, there is an asymmetry: only live neurons are subject to noisy updates. (b) In contrast, Gaussian noise does not share the same assymmetry as SGD noise and is much less prone to dead neuron accumulation (Gaussian noise can revive neurons, contrary to SGD noise). (c) Standard SGD. Dead neurons accumulate quickly in noisy settings, but they plateau when the NN converges (leading to zero gradient). Results are averaged over 3 seeds.

(He et al., 2016) on CIFAR-10 (Krizhevsky et al., 2009) with the Adam optimizer (Kingma & Ba, 2015), with various learning rates and different choices of activation functions. We use a negative slope of $\alpha = 0.05$ for Leaky ReLU, and a $\beta = 1$ for Swish.

Results are shown in Fig 1. We observe a sudden sharp increase in the number of inactive neurons at the beginning of training; few of these recover later in training (see Appendix C). Overall, this leads a significant portion of the 3904 neurons/filters in the convolutional layers of the ResNet-18 to die during training, especially with a high learning rate. Note that this phenomenon is not specific to ReLU activations.

**Intuition.** Similar to Maxwell's demon thought experiment (Maxwell, 1872), one can picture a playful being, ReLUcifer, overseeing a boundary in the weight space that demarcates active and inactive neuron regions. Neurons can move freely within the active zone, but entering the inactive region – where all movement is impeded – is a one-way process governed by ReLUcifer. If the neuron's movements include random components, a risk of inadvertent crossover appears. This risk would be influenced by various factors: noise from the data, being too close to the border, and taking imprudent gradient steps that are too large. Once in the inactive zone, neurons can only be reactivated if the boundary itself shifts. This asymmetry makes it more likely for neurons to die than to revive.

This analogy can be formalized as a biased random walk, an exercise that we touch upon in Appendix B. It also motivates further exploration into how the stochastic nature of various optimizers – related in particular to learning rate and batch size (He et al., 2019; Smith et al., 2018) — contributes to the accumulation of dead neurons in neural networks.

**Role of noise.** Although not all saturated units are due to noise, an important question is how much can noise contribute to neuron death. We argue that noisy training can significantly contribute to dead neuron accumulation. To verify that noise in itself is enough to kill neurons, we trained a 3 layers-deep MLP (of size 100-300-10) on a subset of 10 000 images of MNIST dataset. To isolate the noise from a minibatch (of size 1) gradient ($\hat{g}(\boldsymbol{w}_j^t)$) we deduced from it the full gradient ($g(\boldsymbol{w}_j^t)$).

Figure 2 shows that noise can indeed contribute to dead neuron accumulation and that we should not expect that every neuron dying during training did so because of their individual gradient pointing toward the dead region. We also compare with different noisy regimes to illustrate that the noise structure of SGD plays an important role in the final amount of dead neurons.

### 3.2 TRAINING HYPERPARAMETERS IMPACT ON DYING RATIOS

We close this section by testing empirically some of the implications of the above discussions. The main goal is to quantify the impact of hyperparameters on the ratio of dead neurons. The setup is the same as in Section 3.1. Additional training details can be found J.1.

**Learning rate and batch size.**  Our simple model expose a link between learning rate, batch size, and dying probability: by influencing the noise variance of the optimizer updates, they both should impact the ratios of dying neurons. This prediction proves accurate, as depicted in Fig. 3.

**Regularization.**  Regularization is a popular strategy to control the volume of the solution space that ML optimizers can reach. It restrains the model capacity by favoring solutions with smaller norms, i.e. solutions that are closer to the origin. We remark that for a NN having ReLU activations, $\boldsymbol{w}_j^t = \boldsymbol{0}$ is a point that belongs to the dead region, likewise for the points where all parameters of a neuron are negative, ensuring $\mathrm{ReLU}(\boldsymbol{w}_j \boldsymbol{x}_l^T) = 0$ (since $x_l^i >= 0$, that is the layer inputs are always positive in ReLU networks).

Even if we do not know where the actual death border lies in parameter space, getting closer to the origin is expected to bring a neuron closer to it. According to our model, the neuron should become more likely to die by doing so. As such, regularization can also be an important factor influencing dead neuron accumulation, as empirically demonstrated in Fig. 10.

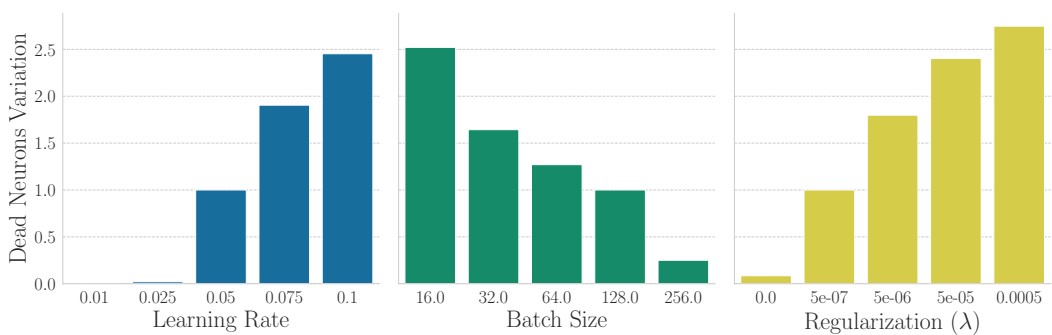

Figure 3: Varying the hyperparameters of a ResNet-18 (CIFAR-10) impacts the number of dead neurons. For the learning rate and batch size histogram, we varied around the combination learning rate 0.05, batch size 128 and λ=0, which on average led to 388 neurons at the end of training. The bar heights indicate the multiplicative ratio of dead neurons with respect to this base configuration. For regularization, we started with learning rate 0.005, batch size 128 and $\lambda = 5 \times 10^{-7}$ (1257 final dead neurons). For the batch size variation, we kept the number of training steps **constant** for a fair comparison. Quantities are averaged over 3 random seeds.

**Optimizer.**  The choice of optimizer inevitably influences the final count of dead neurons post-training, by altering the effective learning rate per parameter. We observed a notable discrepancy when using the ADAM optimizer (Kingma & Ba, 2015) as opposed to SGD with momentum (refer to figure 10). As also highlighted by Lyle et al. (2023), we hypothesize that this discrepancy is primarily attributed to the specific selection of hyperparameters for the ADAM optimizer ($\beta_1, \beta_2, \epsilon$), which significantly impacts neuron death. We further discuss this in Appendix E.

## 4 PRUNING METHOD

The observations collected have a direct application for structured pruning: removing the dead neurons arising during the training process. This simple pruning criterion comes with the main advantage of requiring no additional overhead for its implementation. It only requires monitoring the activation outputs, already computed by the forward pass during training.

From the previous section, we know that neuron sparsity can be influenced by the learning rate, the batch size, the optimizer, and regularization strength. However, the optimizer choice is usually a design choice, while varying the learning rate and the batch size can cause instability during optimization (Cohen et al., 2021). Moreover, performing a grid search over all those hyperparameters would be costly, defeating the purpose of pruning the network during training for acceleration purposes. The possibility to do so however remains if the intent is to maximize sparsity at inference. In the rest, for its simplicity and convenience, we decide to resort to controlling the regularization strength as a mechanism to control sparsity. This choice is backed by the works of Wang et al. (2019) and Wang et al. (2021) that demonstrated the potential of L2 regularization for structured pruning. While similar in spirit, there are notable differences between ours and their approaches:

1. Their methods are for doing structured pruning on a pre-trained NN, while the method we propose is for structured pruning during the initial training phase, recovering a sparse NN directly after. As such, the analysis to justify their methods relies on the solution properties at convergence. The justification we provide for our method relies on its observed impact on the training dynamics.

2. Wang et al. (2021) uses L2 regularization to exploit the underlying Hessian information, and they use L1-norm as a pruning criterion. We use regularization to promote neuron death during training and the criterion for pruning is neural activity.

Our approach to pruning intersects with existing criteria, such as saliency (Molchanov et al., 2016) – dead neurons have a null gradient and would be picked up by this criterion. However, there is a significant shift in pruning methodology: our method influences the learning dynamics to learn sparser solutions. The need to score individual neurons for ablation is removed, observing neuron activations during the forward pass is sufficient to recover the generated sparse subnetwork. We named our method Demon's Pruning (DemP), drawing from the analogy that inspired our work and alluding to the method's darker aspect, namely, the anticipation of neural death. DemP is derived from the interplay of a single hyperparameter – regularization – and dead neurons as measured in section 3. The interplay with the other hyperparameters is not leveraged by DemP, leaving space for a broader exploration of new methods that act on the learning dynamics to retrieve sparser solutions. We now describe the specifics of our pruning method.

**Dynamic Pruning.** To realize computational gain during training, we prune dynamically the NN at every $k$ steps (with a default of $k = 1000$). Dead neurons end up removed almost as they appear, not giving them any chance to be revived. This strategy allows to speed up the training with no significant change in performance (see Appendix H.1). Additionally, it removes the need for choosing correctly *when* to prune Wang et al., 2021; Rachwan et al., 2022; neurons die by themselves during training and *can be removed safely without degrading the current and future performance* (Fig. 13). The need for iterative pruning (Verdenius et al., 2020) also becomes unnecessary since pruning is not done in a single shot anymore, but instead happens gradually during the early training phase. We note that this smooth gradual pruning process is compatible with our approach in part because there is no added cost for computing the pruning criterion.

**Dead Criterion Relaxation.** The definition we choose for a dead neuron asks for it to be inactive to the entire dataset. In practice, we found that this criterion could be relaxed and defaulted to using 1024 examples from the training dataset to measure the death state (making the measurement across multiple minibatches when necessary). Fig. 14 shows that using this proxy for tracking dead neurons is sufficient.

**Regularization Schedule.** Because we noticed that neurons tend to die in the early phase of training, we gradually decay the regularization parameter over the course of training, possibly allowing the remaining neurons to recover from the earlier high regularization. Empirically, we found that using a one-cycle scheduler for the regularization parameter ($\lambda$) is a good strategy (Appendix H.3).

**Weight Decay.** Our method defaults back to traditional regularization, with a term added directly to the loss, as opposed to the weight decay scheme proposed by Loshchilov & Hutter (2019). By doing so, the adaptive term in optimizers takes into account regularization, and neurons move more quickly toward their death border. From a pruning perspective, it allows to achieve higher sparsity than weight decay for the same regularization strength. This is desirable because regularization affects noticeably the performance at high values.

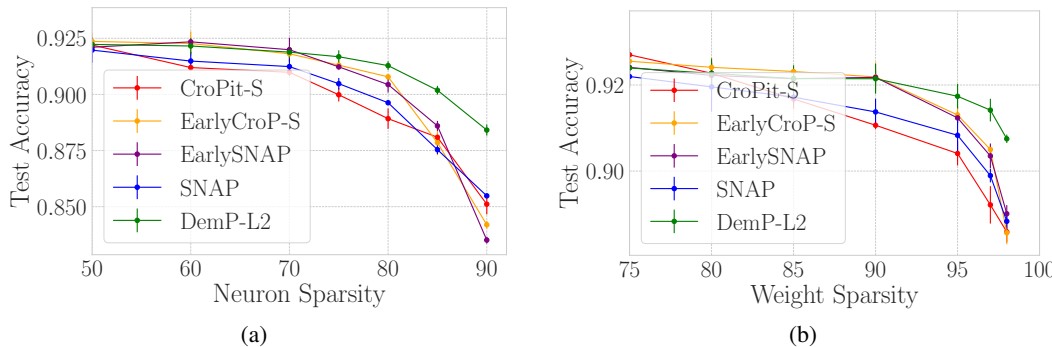

Figure 4: For ResNet-18 networks on CIFAR-10 trained with ADAM, DemP can find sparser solutions maintaining better performance than other structured approaches. **Left:** Neural sparsity, structured methods. **Right:** Weight sparsity, structured methods.

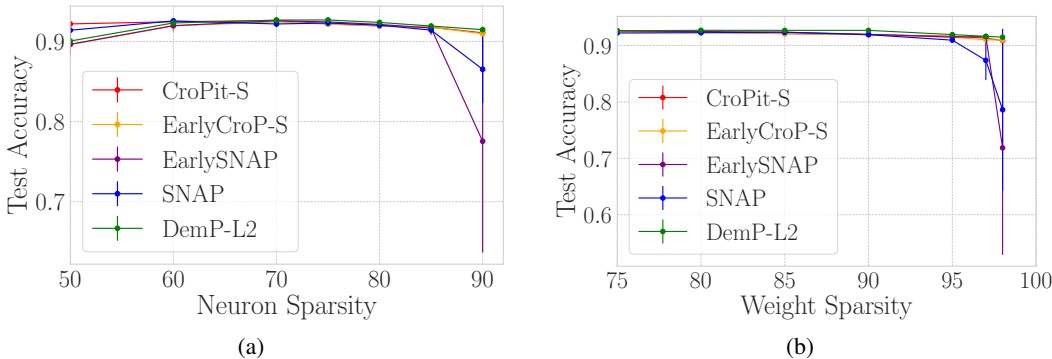

Figure 5: The results on VGG-16 networks trained with ADAM on CIFAR-10. DemP better maintains performance at higher sparsities than other structured approaches. **Left:** Neural sparsity, structured methods. **Right:** Weight sparsity, structured methods.

## 5 EMPIRICAL EVALUATION

We focus our experiments on computer vision tasks, which is standard in pruning literature (Gale et al., 2019). We train ResNet-18 and VGG-16 netowrks on CIFAR-10, and ResNet-50 networks on ImageNet (He et al., 2016; Simonyan & Zisserman, 2015; Krizhevsky et al., 2009; Deng et al., 2009). We follow the training regimes from Evci et al. (2020) for ResNet architectures and use a setting similar to Rachwan et al. (2022) for the VGG to broaden the scope of our experiments. More details are provided in Appendix J.

Our method is a structured one, removing entire neurons at a time. The pruning happens during training, going from a dense network to a sparse one. The methods we compare with also fall into this paradigm, excluding methods like Lasby et al. (2023) which achieves impressive performance, but remains essentially unstructured pruning followed by structured reorganization. We employ the following structured pruning baselines: Crop-it/EarlyCrop (Rachwan et al., 2022), SNAP (Verdenius et al., 2020) and a modified version using the early pruning strategy from Rachwan et al. (2022) (identified as EarlySNAP). The baselines were trained using the recommended configuration of the original authors, and are not subjected to the regularization schedule employed by our method. In all scenarios, our method matches or outperforms those other structured pruning methods (Fig. 4, 5, 6, 7, and Table 1).

We included results from Lee et al. (2023) and Evci et al. (2020) in Table 1 to better illustrate the trade-off between structured and unstructured pruning methods. While unstructured methods currently offer more potential to maintain performance at higher parameter sparsity, structured methods offer direct speedup advantages.

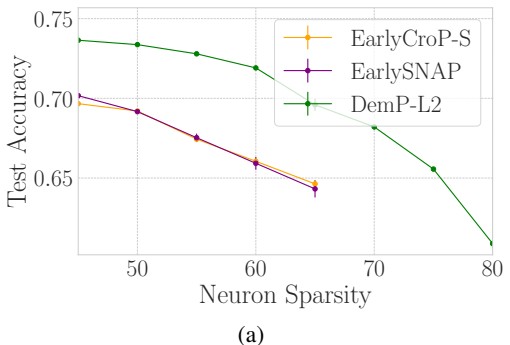 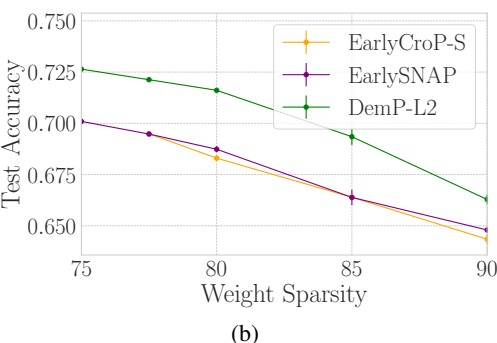

(a)                                    (b)

Figure 6: DemP also outperforms other structured approaches for ResNet-50 networks trained with ADAM on ImageNet, identifying more neurons that can be removed without degrading performance. SNAP and CroPit-S are excluded since they underperform considerably in this setting (see Table 1). **Left:** Neural sparsity, structured methods. **Right:** Weight sparsity, structured methods.

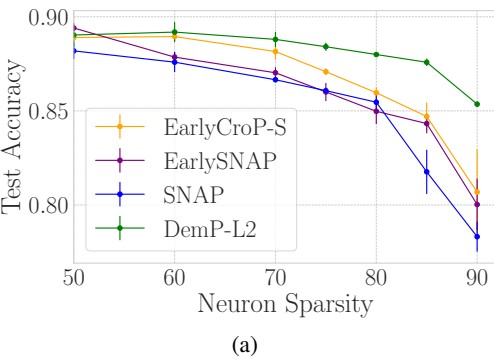 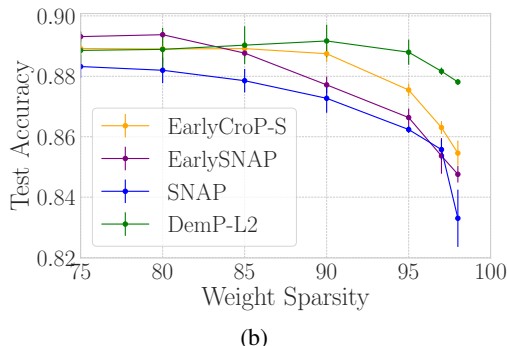

(a)                                    (b)

Figure 7: ResNet-18 networks with *Leaky ReLU* trained on CIFAR 10. DemP again outperforms the baseline structured pruning methods. **Left:** Neural sparsity, structured methods. **Right:** Weight sparsity, structured methods.

**Leaky ReLU.** Dead neurons, and thus the pruning mechanism behind our method, are naturally defined with ReLU activation functions, in which neurons can completely deactivate. However, multiple activation functions, such as Leaky ReLU (Maas et al., 2013), also exhibit a "soft" saturated region. We postulate that neurons firing solely from the saturated region do not contribute much to the predictions and can be considered *almost dead*. We test this hypothesis by employing our method in a network with Leaky ReLU activations (Fig. 7), removing neurons with only negative activation across a large minibatch. Again, our method is able to outperform other structured methods.

## 6 CONCLUSION

In this work, we have revealed how stochasticity can lead to sparsity in neural networks optimized with SGD-like methods. We have empirically demonstrated—and elucidated intuitively—how factors such as learning rate, batch size, and regularization, along with architectural and optimizer choices, collectively impact the sparsity of trained neural networks by influencing the number of neurons that die throughout the learning process. We highlighted that such effects, indicative of a loss in plasticity, can paradoxically be advantageous in a supervised learning setting, contrasting sharply with continual and reinforcement learning settings where they are deemed detrimental. Exemplifying this, we showed how the relationship between regularization and dead neurons can be leveraged to devise a simple yet effective pruning method.

This simplicity makes us confident to be able to adapt the method to a variety of situations. To make it compatible with settings specifying the desired level of sparsity in advance, we could continuously

Table 1: Comparison between different criteria when pruning a ResNet-50 trained on ImageNet around 80% (first line) and 90% (second line) weight sparsity. Because structured pruning methods do not have precise control of weight sparsity, we reported the numbers closest to these target values that we have obtained. $\pm$ indicates the standard deviation, computed from 3 seeds for the structured methods. The sparsity numbers indicate the removed ratio.

| | Method | Test accuracy | Neuron sparsity | Weight sparsity | Training speedup | Training FLOPs | Inference FLOPs |
|---|---|---|---|---|---|---|---|
| | Dense | 74.98% $\pm 0.08$ | - | - | 1.0x | 1.0x (3.15e18) | 1.0x (8.2e9) |
| Structured | SNAP | 28.28% $\pm 0.08$ | 36.9% | 81.4% | 0.51x | 0.32x | 0.32x |
| | | 27.17% $\pm 0.07$ | 56.0% | 90.1% | 0.48x | 0.25x | 0.25x |
| | CroPit-S | 28.34% $\pm 0.52$ | 36.9% | 81.4% | 0.52x | 0.32x | 0.32x |
| | | 27.36% $\pm 0.16$ | 53.2% | 89.9% | 0.47x | 0.27x | 0.27x |
| | EarlySNAP | 68.67% $\pm 0.15$ | 51.70% | 80.37% | 0.95x | 0.63x | 0.63x |
| | | 63.80% $\pm 0.58$ | 66.6% | 90.06% | 0.75x | 0.46x | 0.45x |
| | EarlyCroP-S | 68.26% $\pm 0.31$ | 51.60% | 79.97% | 0.94x | 0.66x | 0.66x |
| | | 64.20% $\pm 0.27$ | 66.6% | 90.37% | 0.82x | 0.51x | 0.50x |
| | DemP-L2 | **71.52%** $\pm 0.09$ | 61.83% | 80.13% | 0.81x | 0.57x | 0.49x |
| | | **66.34%** $\pm 0.16$ | 74.1% | 89.93% | 0.61x | 0.42x | 0.34x |
| Unstructured | Dense[†] | 76.67% | - | - | - | - | - |
| | Dense* | 76.8 $\pm 0.09$ % | - | - | - | 1.0x (3.2e18) | 1.0x (8.2e9) |
| | Mag[†] | **75.53%** | - | 80% | - | - | - |
| | Sal[†] | 74.93% | - | 80% | - | - | - |
| | SET* | 72.9% $\pm 0.39$ | - | 80% | - | 0.23x | 0.23x |
| | | 69.6% $\pm 0.23$ | - | 90% | - | 0.10x | 0.10x |
| | RigL (ERK)* | 75.10% $\pm 0.05$ | - | 80% | - | 0.42x | 0.42x |
| | | **73.00%** $\pm 0.04$ | - | 90% | - | 0.25x | 0.24x |

[†] values obtained from Lee et al. (2023)
* values obtained from Evci et al. (2020)

increase regularization before cutting it off at the target ratio. It is also easy to extend existing pruning methods with it. Multiple pruning criteria will better identify the parameters to prune if they belong to a dead neuron. Unstructured methods could leverage the added structure sparsity of high regularization to achieve better computational gains.

Moreover, our experiments with Leaky ReLU exemplify that the methodology is compatible with activation functions that feature a softer saturation region than ReLU. This opens up the possibility to sparsify transformer architectures (Vaswani et al., 2017) during training since they commonly rely on activation functions such as GELU and Swish. Due to the model sizes involved in their typical training regimes, the computational gains and environmental benefits of applying our methodology there could be considerable.

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
