## A   DEAD NEURONS IN CONVOLUTIONAL LAYERS

In convolutional layers, ReLU is applied element-wise to the pre-activation feature map. We consider an individual neuron (filter) dead if all elements of the feature map post-activation are 0. Formally,

**Definition:** The $j$-th neuron/filter in the convolutional layer $\ell$ is *inactive* if it consistently outputs a feature map (post-activation) with elements summing to zero on the entire training set, i.e $\sum_{k,l} F^\ell_{jkl} = 0$. A neuron/filter that becomes and remains inactive during training is considered as *dead*.

## B   BIASED RANDOM WALK MODEL

To formalize the intuition from section 3.1, we follow a standard line of work (Cheng et al., 2020) taking the view of SGD in Eq. 1 as a biased random walk (Anderson, 1998), described by the Langevin process,

$$\boldsymbol{w}_{t+1} = \boldsymbol{w}_t - \eta g(\boldsymbol{w}_t) + \sqrt{\eta}\,\hat{\boldsymbol{\xi}}(\boldsymbol{w}_t, b_t) \tag{2}$$

where the zero mean variable $\hat{\boldsymbol{\xi}}(\boldsymbol{w}, b) := \sqrt{\eta}\,(g(\boldsymbol{w}) - \hat{g}(\boldsymbol{w}, b))$ represents the gradient noise. In the limit of small learning rate, Eq. 2 is well approximated (Cheng et al., 2020, Theorem 2) by the following stochastic differential equation (SDE),

$$d\boldsymbol{w}_t = -g(\boldsymbol{w}_t) + M(\boldsymbol{w}_t)d\boldsymbol{B}_t, \tag{3}$$

where $\boldsymbol{B}_t$ denotes a standard Brownian motion and $M(\boldsymbol{w}) := \sqrt{\mathbb{E}_b[\hat{\boldsymbol{\xi}}(\boldsymbol{w}, b)\hat{\boldsymbol{\xi}}(\boldsymbol{w}, b)^\top]}$.

A key aspect of the gradient noise in SGD is that it is *multiplicative*, i.e., it depends on the parameter. Now, a well-known property of systems with multiplicative noise is that regions of lower noise magnitude can act as attractors (Oksendal, 2010). Intuitively, this is because the noise pushes the system away from regions where it has a higher impact, leading to a higher probability of staying in regions where it has a lower impact. Mathematically, this is characterized by a tendency for the invariant distribution to have higher probability density in regions of lower noise magnitude.

We illustrate this on a simplified version of Eq. 3, which partially captures the effect we want to highlight.

**Absorbing Brownian motion**. We consider a one-dimensional absorbing Brownian motion with a boundary at zero, which can be described by the SDE:

$$dw_t = \begin{cases} \sqrt{\eta}dB_t & \text{as long as } w_t > 0 \\ 0 & \text{otherwise} \end{cases} \tag{4}$$

It models a system subject to noise in an 'active' region $w > 0$, which gets stopped at 0 – and remains there, hence 'dies' – once it hits 0. This illustrative example can be thought of as a simplified description of a regime where the dynamics (4) is dominated by noise, such as e.g., a neuron encoding features with very low correlation to the task.

In this case, the probability that the system is still active at time $t$, i.e $w_t > 0$, is related to the distribution of the first hitting time at 0 of a standard Brownian motion: it is given by $P(T_0 > t)$, where $T_0 = \inf\{t \geq 0 : B_t = 0\}$. A well-known property of Brownian motion (Karatzas & Shreve, 2014) is $\lim_{t\to\infty} P(T_0 > t) = 0$, which shows that the system Eq. (4) eventually dies with probability 1. More generally, the following result specifies the dependence on initialization:

**Proposition 1.** *Consider the system 4 initialized at $w_0 > 0$. Then the probability that the system is still active at a given time $t > 0$ is given by*

$$P(w_t > 0 \,|\, w_0) = \sqrt{\frac{2}{\pi}} \int_0^{w_0/\sqrt{t}} e^{\frac{-u^2}{2\eta}}\, du \tag{5}$$

Prop. 1 implies that $(i)$ the system eventually dies almost surely, $(ii)$ for any given finite horizon time $t$, the smaller the initialization, the more likely the system is dead at $t$, $\lim_{w_0\to 0^+} P(w_t = 0 \,|\, w_0) = 1$. Finally, the dependence on the scaling $\eta$, which represents the learning rate, illustrates how a noisier environment can accelerate this dying process.

## C  FEW DEAD NEURONS REVIVE

While empirical observations have shown a gradual accumulation of dead neurons (Fig. 1), we also observed that neurons can revive (Appendix D.1). To better assess the potential impact of reviving neurons on performance, we measured the overlap ratio ($|X \cap Y| / \min(|X|, |Y|)$) between the historical set of dead neurons at previous iterations and the set of dead neurons at the current iteration. This methodology directly follows Sokar et al. (2023). The results in figure 8 show that most neurons (over 90%) inactive at any point during training end up dead at the final iteration. This – coupled with our results showing that dying neurons can be dynamically pruned during training without impacting performance (Appendix H.1) – strongly suggests that neurons becoming inactive at any point during training in ReLU networks do not contribute significantly to the final performance of the trained model.

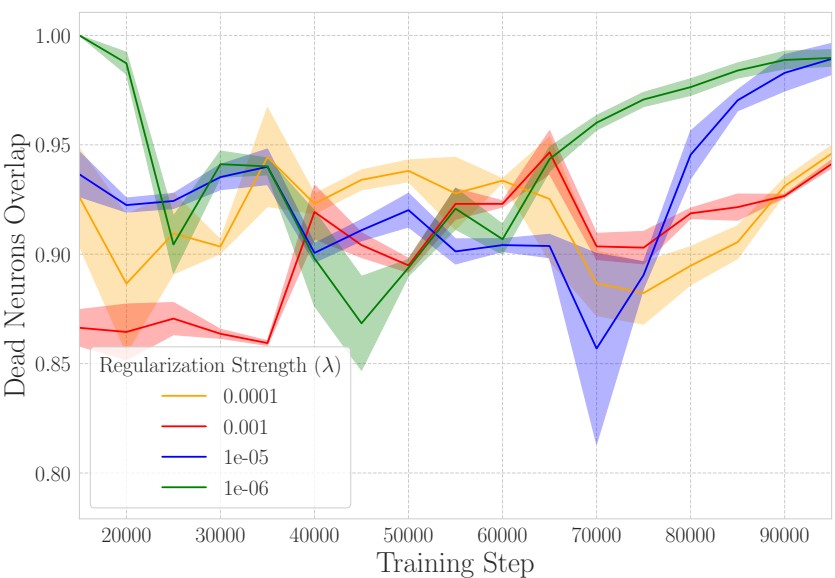

Figure 8: Overlap ratio of dead neurons during training, as measured across all layers of a ResNet-18 trained with ADAM on CIFAR-10 at various regularization strengths. Results are shown for training steps bigger than 15k because no dead neurons were observed previously for the lowest regularization strength. We observe that the vast majority (over 85%) of neurons dying early never revive. More importantly, even if they may have been revived earlier, $\approx 95\%$ of neurons that became inactive at any point during training are dead when training finishes.

## D  HYPERPARAMETERS IMPACT, ADDITIONAL RESULTS

### D.1  TRAINING TIME

The relation with training time, asserting that the probability of a neuron dying increases as training progresses (Prop. 1) doesn't entirely align with practical applications. Modern overparameterized architectures often have the capacity to memorize the entire training dataset, achieving zero loss in the process. Given that the gradient signal is proportional to the loss, it would concurrently diminish to zero for all neurons, preventing any further death.

We observe a pattern consistent with this idea (Fig. 1), where the total count of dead neurons spikes sharply in early training to then fluctuate slightly before stabilizing. The fluctuations demonstrate that neurons *can indeed revive*. However, additional experiments with ReLU networks revealed that most reviving neurons die again later (Fig. 8) and that their dynamic elimination has negligible to no impact on performance (Fig. 13).

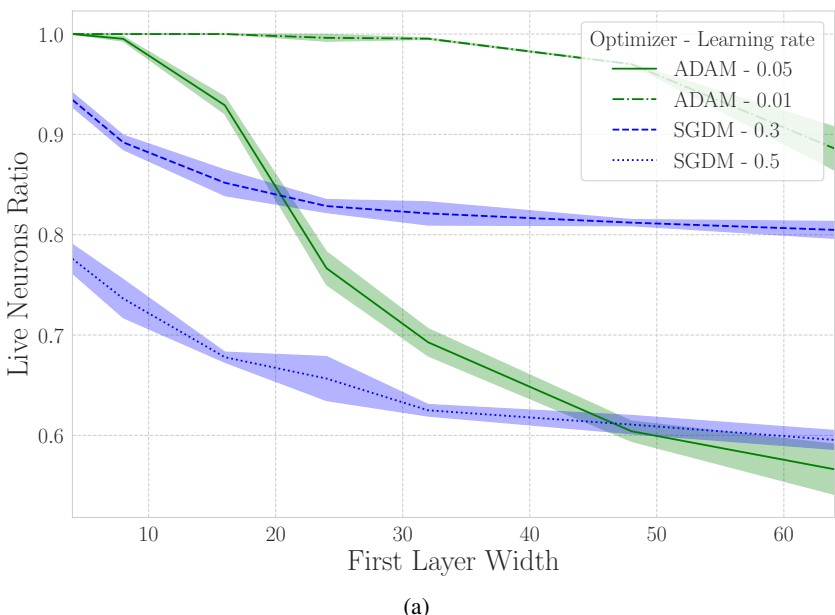

(a)

Figure 9: An increased width leads to a higher ratio of neurons dying, independently of the optimizer. We use the number of channels in the initial layer of the ResNet-18 to indicate the width, with 64 being the typical number of channels in the first convolution layer.

## D.2    NETWORK WIDTH

The widths of a neural network's layers also influence the ratio of live neurons (live neurons to total neurons in the network) post-training (see figure 9). Typically, this ratio increases with the width; however, the total number of live neurons continues to rise with increased width. This phenomenon is somewhat anticipated as incorporating more neurons with random initialization in any given layer can only amplify the training noise, especially in the initial phase. Moreover, since initialization functions usually adjust their standard deviation proportionally to the number of channels ($\sigma \propto \sqrt{\frac{1}{\text{fan\_in} + \text{fan\_out}}}$), widening the network places neurons closer to their death border right from the initialization. The connection between width and dead neurons maintains its significance as neural network sizes are inclined to increase over time with the availability of more computational resources. If this trend persists, the accumulation of dead neurons could potentially become increasingly pervasive.

## D.3    REGULARIZATION

We also experimented with a slightly modified version of L2, that solely regularizes the positive weights. The intuition behind it was that by regularizing only the positive weights, the average pre-activation output of a ReLU network would gradually shift toward negative outputs. We dubbed it the *coup de grâce* L2 normalization (CDG_L2), and describe it further in Appendix F. It proved more aggressive than classical L2 regularization, except when using batch normalization (BN) layers with ADAM optimizer (fig.10 and Appendix G).

The particular impact of normalization layers is interesting. Without regularization and with everything else left equal, BN leads to less dead neuron accumulation (Appendix G). But with regularization, two distinct modalities appear in neuron pre-activations distribution: one centered around 0 and another toward the positive values. Normalization then shifts the mean of this distribution to 0, bringing the modality that was previously close to the origin on the negative (dead) side. Normalization paired with regularization can therefore be a very effective strategy for promoting neuron death.

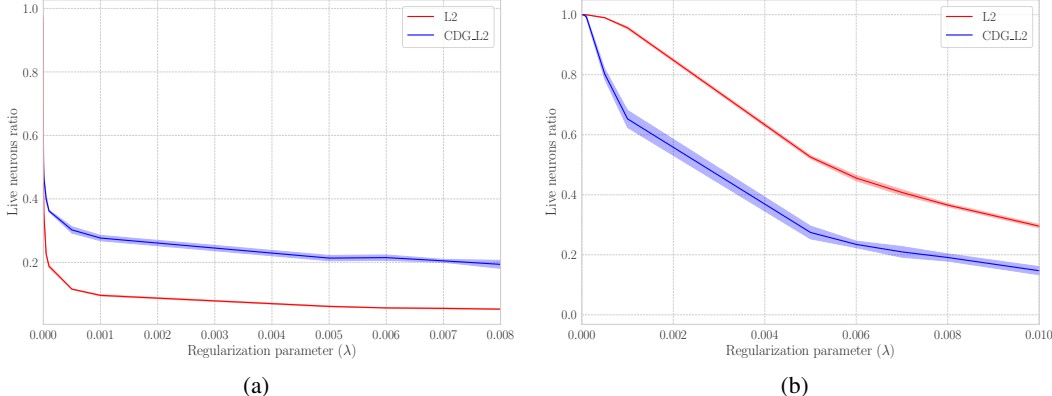

Figure 10: Increasing the regularization parameter ($\lambda$) quickly increases the number of dead units that accumulate during training by bringing the neurons closer to their death border. BN was used in the experiments that generated both figures. **Left**: When using ADAM optimizer, dead neurons accumulate more quickly with classical L2 regularization. ADAM is particularly effective for killing neurons, indicated by the much steeper decrease in live neurons when regularization is increased. **Right:** The situation is reversed when using momentum, with CDG_L2 generating more dead neurons.

## E    ADAM IS A NEURON KILLER

The greater impact of ADAM over the dying ratio compared to momentum must be due to the second-moment term, which is the only significant difference with momentum. Recall that ADAM update is (Kingma & Ba, 2015):

$$m_t = \beta_1 \cdot m_{t-1} + (1 - \beta_1) \cdot g_t$$
$$v_t = \beta_2 \cdot v_{t-1} + (1 - \beta_2) \cdot g_t^2$$
$$\hat{m}_t = \frac{m_t}{1 - \beta_1^t}$$
$$\hat{v}_t = \frac{v_t}{1 - \beta_2^t}$$
$$\theta_{t+1} = \theta_t - \frac{\eta}{\sqrt{\hat{v}_t} + \epsilon} \cdot \hat{m}_t$$

Earlier, we hypothesized that the neurons ending up dead were the ones experiencing very small gradients, such that the noise dominated their update trajectories. If this is the case, $g_t^2$ (the squared gradient) would be very small for those neurons' parameters, eventually leading to a very small second-moment estimation $\hat{v}_t$. In such a scenario, $\epsilon$ would end up dominating $\sqrt{\hat{v}_t}$, effectively multiplying the learning rate by $\epsilon$ which is typically set to $1 \times 10^{-8}$. Moreover, as the decay ($\beta_2 = 0.99$) of $\hat{v}_t$ is usually slower than the one of $\hat{m}_t$ ($\beta_1 = 0.9$), a few sudden noisier updates would be sufficient to make huge random steps.

It is worth noting that RL practitioners typically set epsilon to a higher value (Hessel et al., 2018), as it has empirically been found to perform better. Higher $\epsilon$ values should reduce the number of dead neurons induced by ADAM optimizer, which could be the cause for the improved performance/stability observed in RL. Also, because of constant distribution shifts, rapid accumulation of dead neurons often occurs in RL tasks.

Also notable, HuggingFace Transformers library (Wolf et al., 2020) default $\epsilon$ ADAM parameter to $1 \times 10^{-6}$, following RoBERTa example (Liu et al., 2019). Manipulating the $\epsilon$ parameter of AdaGrad was also observed to impact significantly a transformer performance model (Agarwal et al., 2020). Verifying if those heuristic choices are due to their impact on dead neuron accumulation would be quite interesting.

## F  CDG_L2

Although we don't know where the actual death border lies in a neural network using ReLU, a unit is certain to be dead (except in the initial layer) if all of its weights are smaller than zero, i.e. if $w_j^i <= 0 \; \forall i$. This is because the output of neuron $j$ will then always be negative $\boldsymbol{w}_j \boldsymbol{x}_l^T$ since $x_l^i >= 0$, that is the layer inputs are always positive in ReLU networks.

Therefore, by pushing a neuron's parameters toward the negative side, we should make it more probable to die. It is to that end that we introduce the *coup de grâce* L2 normalization (CDG_L2). Formally, we define it as:

$$CDG\_L2 = \lambda \sum_{w_i > 0} w_i^2 \tag{6}$$

It is in essence L2 regularization applied only to the positive weights of the network. The intuition behind it is that by regularizing only the positive weights, the average pre-activation output of a ReLU network will gradually shift toward negative outputs. This simple change proved enough to push over the dead border neurons that otherwise activated sporadically.

## G  NORMALIZATION LAYERS

To study the impact of normalization layers on dead neurons, we monitored the number of inputs for which individual neurons were activated after training a NN. The experiences were performed with a ResNet-18 trained with ADAM on CIFAR-10, making the range for activation count between 0 and 50,000. This allows us to monitor the amount of dead neurons, the ones for which the activation count is zero. We plotted those results as density histograms. Figure 11 shows the results when using L2 and CDG_L2 with BN layers and in figure 12 we display the results when there are no BN layers in the network. Dead neurons are the ones belonging to the leftmost bucket.

A first observation is that when BN layers are present, there are more neurons activating for all or almost all of the entire dataset. This supports our claim that BN layers can help reduce the amount of dead neurons when everything else is kept equal. But we can also observe that BN layers make the first modality much more light-tailed, meaning that fewer neurons are activated for a small portion of the dataset. We believe this is due to the normalization process that puts the distribution mean at the origin. This brings the entire heavy-tailed, pre-normalized first modality in the negative regions before ReLU. Thus, when BN layers are used, neurons either activate almost all the time or never activate. This in turn makes the dead neuron criterion that we use, i.e. neurons that do not activate for the entire training dataset, particularly appropriate when BN layers are used.

Finally, we also observe that CDG_L2 "drains" the rightmost modality more quickly than L2, supporting our claim that the CDG variant is generally more aggressive for neuron killing.

## H  PRUNING METHOD DETAILS

We validate and justify the heuristic choices made for our pruning method via empirical observation exposed in this section. We used the same setup as before for a ResNet-18 trained on CIFAR-10.

### H.1  DYNAMIC PRUNING

To verify the impact of dynamic pruning, we measured if there were any performance discrepancies when it was enabled or not. Across runs, we varied the regularization strength while measuring accuracy and sparsity. The results, in figure 13, show that enabling dynamic pruning does not affect the final performance. The very slight variations between runs fall well between the expected variance across different runs. This experiment really reinforces the hypothesis that neurons that die and later revive during training do not contribute significantly to the learning process.

### H.2  DEAD CRITERION RELAXATION

To measure if a minibatch could be used to measure the death state instead of the entire dataset, we tracked the number of dead neurons during training with both metrics. We used a minibatch

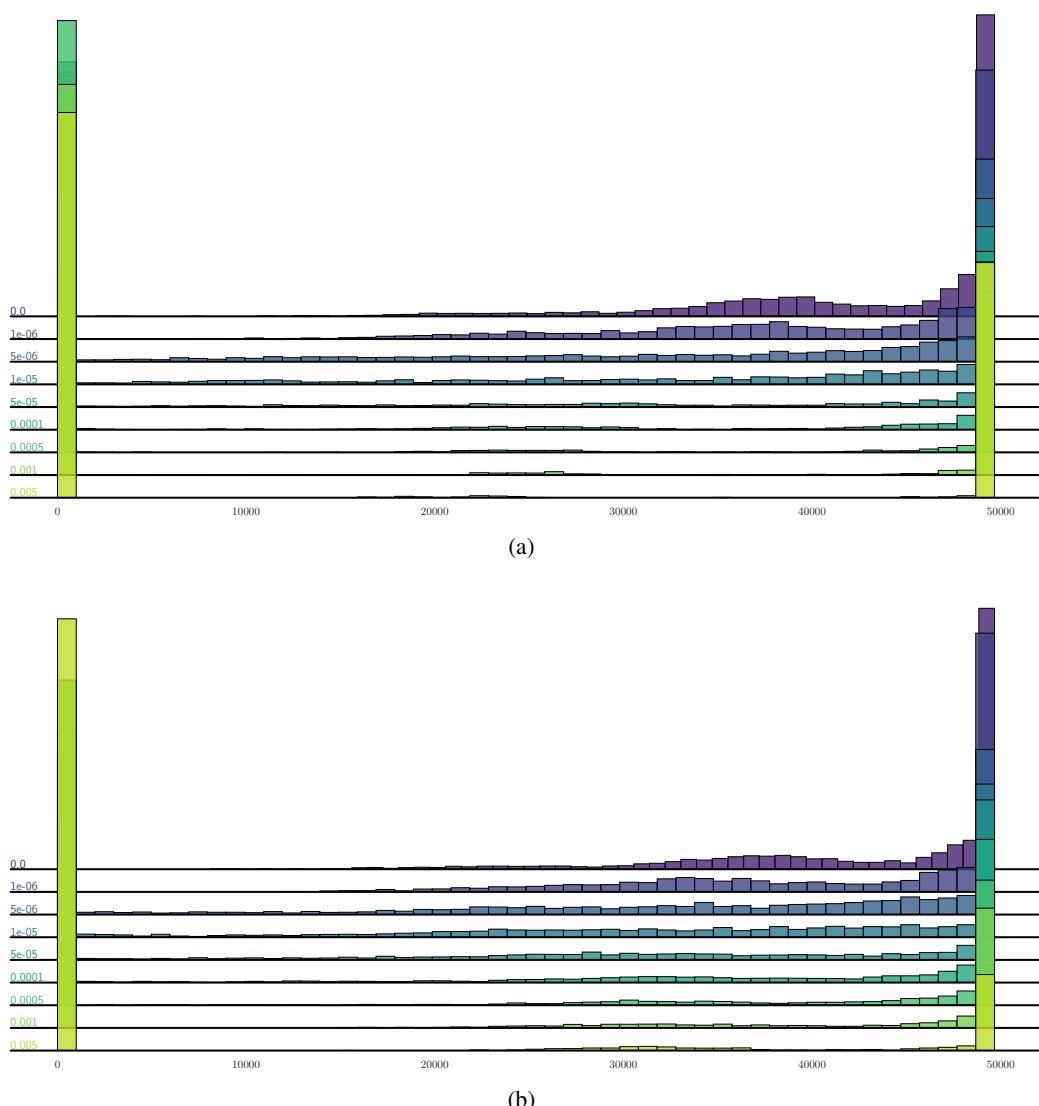

(a)

(b)

Figure 11: Histogram of individual neuron's activation count across the training dataset (CIFAR-10) for a ResNet-18 **with** BN layers. Distributions are shown for different values of the regularization parameter ($\lambda$). Regularization increases toward the front, with the specific value indicated on the far left of the figure. The leftmost modalities are single peaks at 0 when BN layers are present, indicating that neurons either never activate, or activate almost all the time with normalization. **Top:** L2 regularization. **Bottom:** CDG_L2 regularization.

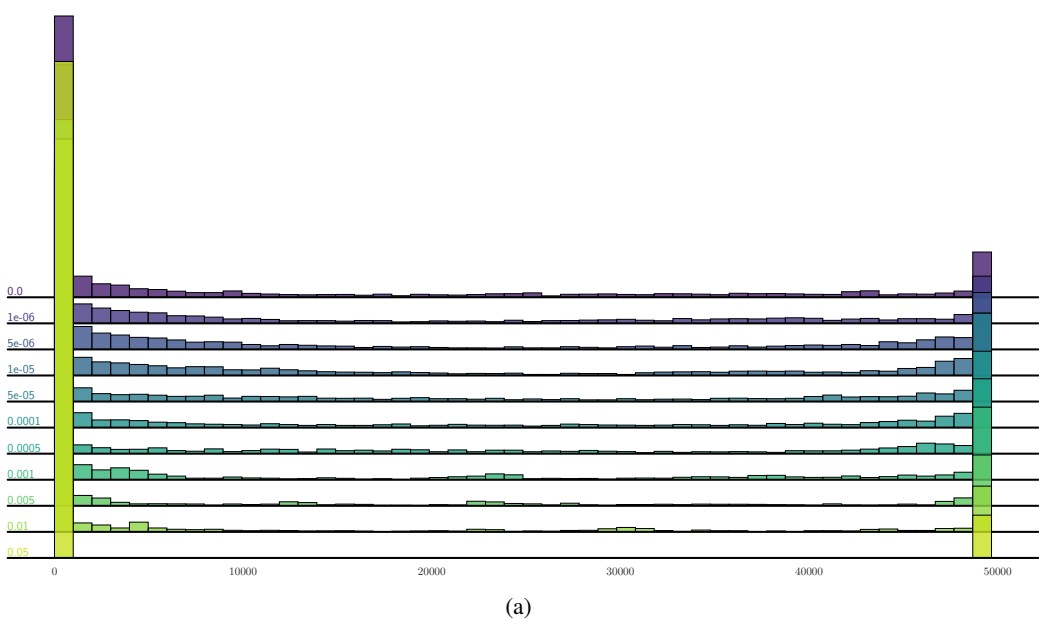

(a)

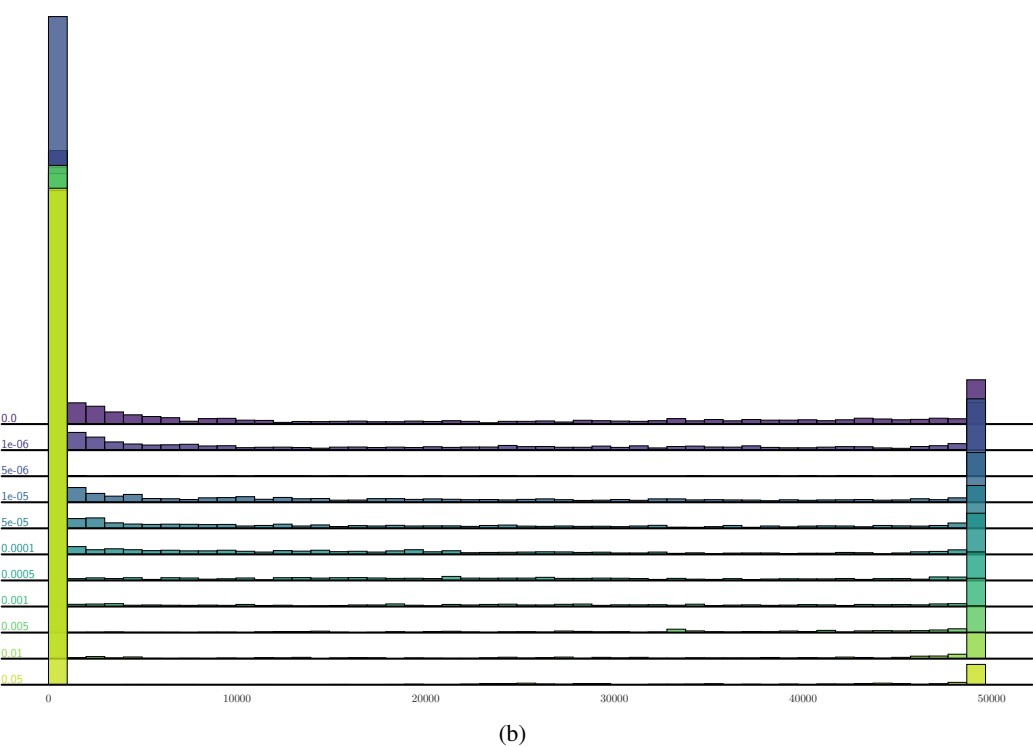

(b)

Figure 12: Same experiments as figure 11, but for a ResNet-18 **without** BN layers. When BN is removed, the leftmost modalities become more heavy-tailed, indicating that a portion of the neurons activate very sparsely across the training dataset. **Top:** L2 regularization. **Bottom:** CDG_L2 regularization.

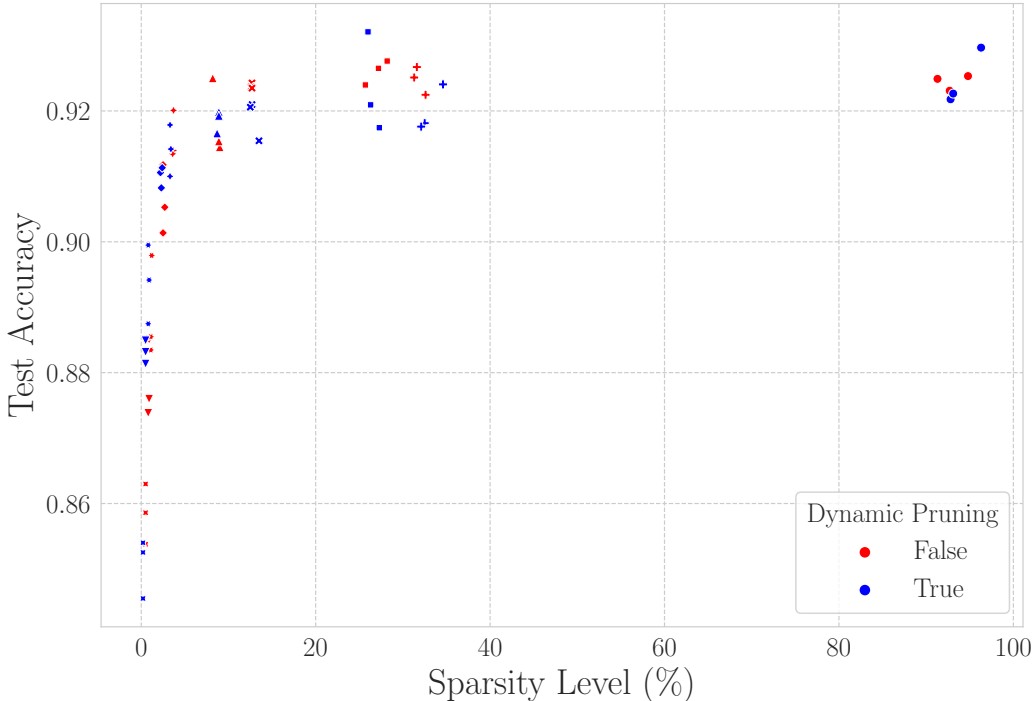

Figure 13: Measuring final accuracy vs. sparsity when dynamic pruning is enabled or not. There are barely any variations between the two strategies, allowing us to conclude that using dynamic pruning does not affect performance. Different symbols were associated to different regularization strengths. Experiment performed with ResNet-18 on CIFAR-10 for 3 seeds.

containing 512 inputs from the training dataset for the proxy measurement. We can see that both curves closely track each other. More importantly, they match at the end of the training, indicating that overall the same amount of neurons would be removed when performing the death check over the minibatch. Dynamic pruning was disabled for this experiment.

### H.3 DECAYING THE REGULARIZATION PARAMETER

Finally, we also empirically tested different schedules over the regularization parameter, trying to mitigate the impact of high regularization by decaying the parameter over the course of the training. We settled on using a one-cycle scheduler for the regularization strength because of slightly better performance in the higher sparsity level. However, we remark that even a constant schedule over the regularization parameter is sound with our method

## I ADDITIONAL COMPARISON WITH UNSTRUCTURED METHODS

We employ the `JaxPruner` package (Lee et al., 2023) to illustrate further trade-off of our method against some unstructured methods. Our method is capable of achieving *similar* performance to unstructured ones for the ResNet-18 and VGG-16 experiments (Fig. 16). The comparisons with the unstructured methods use their default configuration from JaxPruner, which was tuned for a ResNet-50. We expect their performance on ResNet-18 and VGG-16 to be improved by tuning the pruning distribution, the pruning schedule, and the pruning iterations scheme (Lee et al., 2023). However, for those not interested in expensive tuning, our method becomes an interesting default choice.

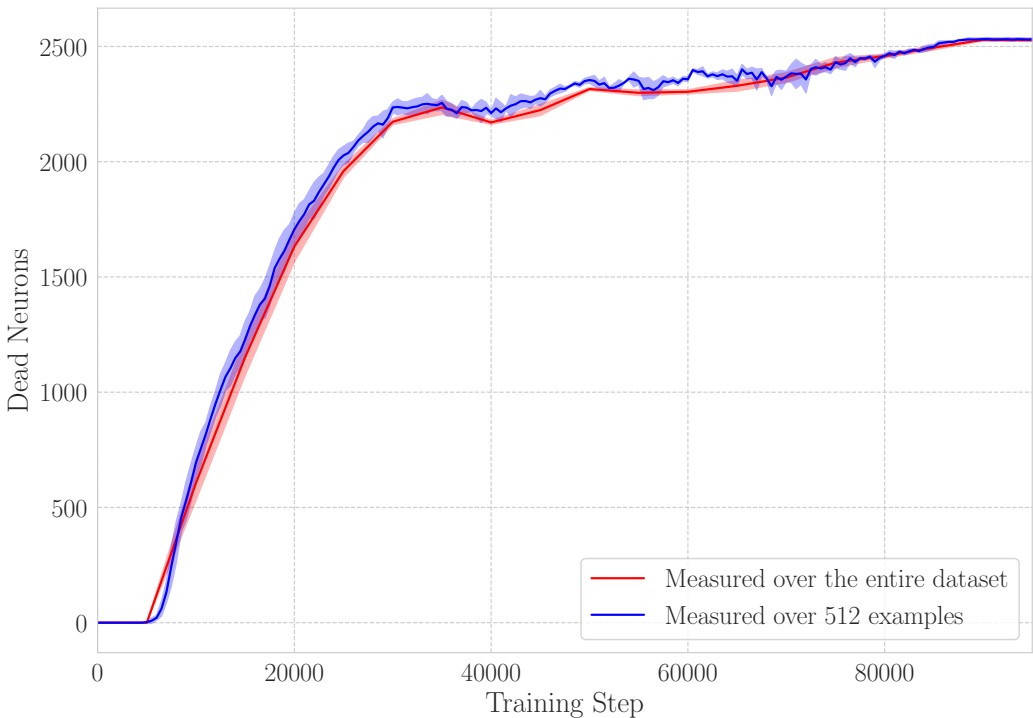

Figure 14: Instead of validating the death state of neurons against the entire training dataset, it proves sufficient to use a smaller dataset. The curves meet at the end of the training, indicating that the same final number of neurons would be removed for both strategies. Experiment performed with ResNet-18 on CIFAR-10 for 3 seeds.

## J    IMPLEMENTATION DETAILS

### J.1    RESNET-18/RESNET-50

We mostly followed the training procedure of Evci et al. (2020) for the ResNet architectures.

**ResNet-18.** We train all networks for 250 epochs using a batch size of 128. The learning rate is initially set to 0.005 and is thereafter divided by 5 every 77 epochs. We use ADAM optimizer because it induces higher sparsity. While varying regularization (L2 or CDG_L2) is used with our method, we default to a constant weight decay (0.0005) for all other methods than ours. Random crop and random horizontal flips are used for data augmentation.

**ResNet-50.** We trained the ResNet-50 for 100 epochs, with a batch size of 256 instead of 4096. The initial learning rate is set to 0.005, before being decayed by a factor of 10 at epochs 30, 70, and 90. Label smoothing (0.1) and data augmentation (random resize to either $256 \times 256$ or $480 \times 480$, before randomly cropping to $224 \times 224$. Followed by random horizontal flip and input normalization) are also used. We again use ADAM, vary the regularization with our method but use a constant weight decay (0.0001) for other methods.

### J.2    VGG-16

We followed a training procedure similar to Rachwan et al. (2022). We used ADAM with a learning rate of 0.005 and a batch size of 256, with the One Cycle Learning Rate scheduler (Smith & Topin, 2018). The networks are trained for 80 epochs. CIFAR-10 images are normalized and resized to $64 \times 64$ before applying random crop and random horizontal flip for data augmentation.

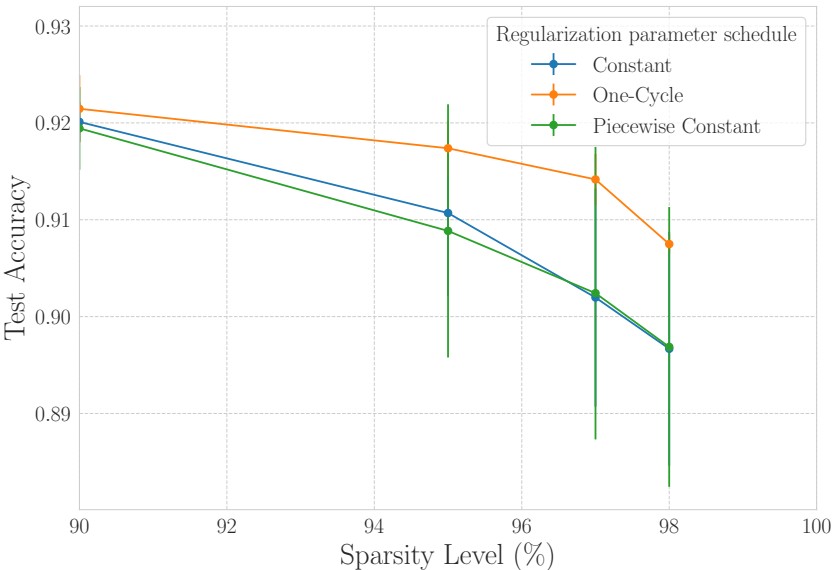

Figure 15: We studied the impact of different schedules over the regularization parameter for our method, settling down on a one-cycle scheduler as default. Experiment performed with ResNet-18 on CIFAR-10 across 3 seeds.

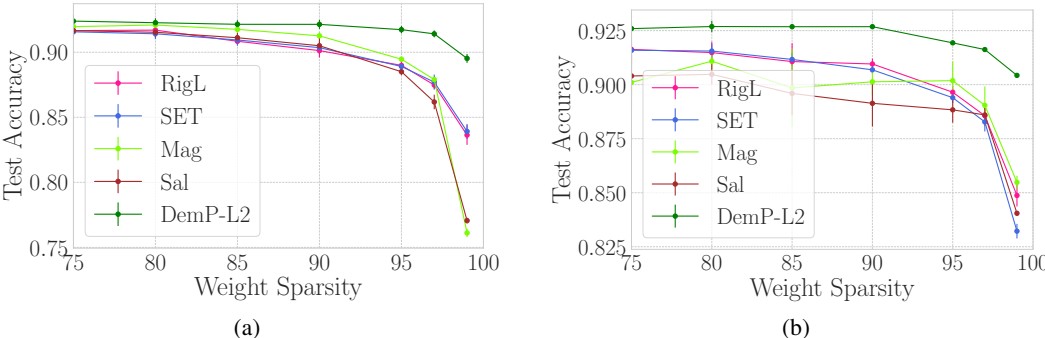

Figure 16: At default configuration DemP competes with common unstructured methods for networks trained on CIFAR-10, especially at high sparsity. **Left:** Weight sparsity in a ResNet-18. **Right:** Weight sparsity in a VGG-16.

## J.3 STRUCTURED METHODS

We closely reimplement in JAX (Bradbury et al., 2018) the structured methods from Rachwan et al. (2022), keeping all the hyperparameters specific to every method as is. The training hyperparameters are the same as specified in J.1 and J.2.

## J.4 UNSTRUCTURED METHODS

For the unstructured methods, we rely on Lee et al. (2023) implementations, using their method's configuration for pruning a ResNet-50 for all our experiments. The training hyperparameters are the same as specified in J.1 and J.2.