# OpenReview forum: "A Demon at Work: Leveraging Neuron Death for Efficient Neural Network Pruning"
_ICLR.cc/2024/Conference — Submitted to ICLR 2024_

### Official Review · Reviewer_tiWA · 2023-10-27

**Soundness:** 3 good
**Presentation:** 2 fair
**Contribution:** 2 fair
**Rating:** 5
**Confidence:** 2

**Summary:**

This paper introduces "Demon's Pruning" (DemP), an innovative method that utilizes dying neurons for model optimization, and provides a comprehensive exploration of hyperparameters influencing neuron mortality. The authors demonstrate DemP's superior performance over existing structured pruning techniques and its competitive results with unstructured methods.

**Strengths:**

1. Insightful exploration of dying neurons, establishing their utility in structured pruning algorithms.
2. Introduction of the simple yet broadly applicable DemP, which outshines current structured pruning techniques.
3. Extensive empirical validation of DemP's effectiveness across multiple benchmarks, with comparative analysis against other pruning methods.

**Weaknesses:**

1. The paper's motivation is somewhat unconvincing. Encouraging neuron death during training may compromise the expressivity of neural networks, potentially leading to performance degradation. The authors, however, propose accelerating neuron death and subsequent pruning. What is the motivation to accelerate the neuron death? (I understand the authors' motivation, i.e., pruning, but why just have a "normal" speed of pruning as the neurons will die at the end of training with a high probability.) Besides, It's worth noting that several existing works already prune networks by removing small-weight "dead" neurons without the need for prompting [1, 2] in structured pruning and unstructured pruning, e.g., magnitude.
2. the paper lacks direct evidence that dead neurons **remain inactive** during training, despite the existence of previous works validating this hypothesis, such as the overlap coefficient [3].
3. The experiments in Fig 2 suggest that noise may contribute to the accumulation of dead neurons, but its role appears to be more like an amplifier than a key driver. Moreover, the effect seems to depend on the noise type. The section seems more focused on expanding the word count which makes this paper inconsistent, and the experimental details on noisy updating are insufficient. Additionally,

Minor:
1. The unit of Dead Neurons Variation in Fig. 3 is unclear.

References:
1. Sokar, G., Agarwal, R., Castro, P. S., & Evci, U. (2023). The dormant neuron phenomenon in deep reinforcement learning. _arXiv preprint arXiv:2302.12902_.
2. Wang, H., Qin, C., Zhang, Y., & Fu, Y. (2020). Neural pruning via growing regularization. _arXiv preprint arXiv:2012.09243_.
3. Liu, Z., Li, J., Shen, Z., Huang, G., Yan, S., & Zhang, C. (2017). Learning efficient convolutional networks through network slimming. In _Proceedings of the IEEE international conference on computer vision. (pp. 2736-2744).

**Questions:**

see the weakness.

---

> ### Author Response · Authors · 2023-11-21
> **Response to Reviewer tiWA**
>
> We thank the reviewer for the accurate summary of our work and their positive feedback on the soundess and usefulness of our approach and the simplicity and  effectiveness  of our method. We also appreciate their concrete suggestions to better articulate the motivation of our approach and to strengthen some of claims with further evidence.
>
> ************************Weaknesses:************************
>
> 1.  **On the motivation for accelerating neural death**
>
> Thank you for pointing out this confusing aspect of our paper. The motivation to accelerate and promote neural death is threefold:
>
>  *  It alleviates the need for a score function and pruning intervention because the learning dynamics reach directly a sparser solution. It is also a notable difference from previous existing works. We now discuss further this point in the introduction.
>  * Actively promoting neural death and dynamically removing the dead units can lead to training speedup. We added training speedup results in Table 1 to better highlight this aspect.
>  * Because the updates are proportional to the loss, the update size diminishes as the model converges, significantly affecting the dying speed. Actively promoting neural death ensures that dying occurs in a fixed number of times steps. We added Appendix D1 to clearly expose this aspect.
>
> 2. **Evidence that inactive neurons remain inactive**
>
> Thank you for pointing out this omission and the great suggestion. As per  the reviewer’s suggestion, we complemented our previous results justifying dynamic pruning (now Appendix H1) with measurements of the overlap ratio in a new Appendix C (Fig. 8). We have added a reference to the latter to section 3.1.
>
> 3.  **On the role of noise**
>
> Thank you for highlighting the apparent inconsistency. Section 3.1 introduces — and tries to provide intuition behind — the driving hypothesis of our paper: Does the noise level during optimization affect the solution sparsity via dead neuron accumulation?  Section 3.1 and Figure 2 show that SGD noise structure has the potential to amplify dead neuron accumulation, motivating further investigations. The comparison with Gaussian noise highlights the asymmetric difference between the two noise structures, an important aspect that is also part of our thought experiment.
>
> We reworked thoroughly the intuition in section 3.1 and brought further clarification in the Figure 2 caption as well to better justify this section.
>
>
> ******Minor:******
>
> 1. Thank you, we clarified Figure 3.

---

> > ### Comment · Reviewer_tiWA · 2023-11-22
> > **Official Comments from Reviewer tiWA**
> >
> > Thanks for the response! Most of my questions are efficiently addressed. However, regarding motivation: I still hold the belief that neuron death is a trade-off in sparse training and performance improvement because the plasticity loss will incur the decreased performance.
> >
> > In other words, you can not accelerate the neuron death and maintain the performance over the training process at the same, where I also found an interesting paper that maintains this intuition in submission of this conference, but with some interesting techniques to ensure performance improvement during the sparse training from the perspective neuron deaths. I would like to increase to 5 points but a lower confidence.

---

> ### Author Response · Authors · 2023-11-22
> **Further discussion with Reviewer tiWA**
>
> Thank you for your thoughtful reconsideration and adjustment of our score.
>
> We want to emphasize our agreement with the reviewer's point about the inherent sparsity-performance tradeoff. Achieving a favorable balance, where high sparsity levels are attained with minimal impact on performance, is a central challenge faced by all pruning methods. Our paper extensively evaluates and compares these tradeoffs, as illustrated in Figures 4-7. Our results demonstrate a more advantageous tradeoff for our method compared to the strongest competitors in the recent  literature of structured pruning, despite the simplicity of our pruning criterion.
>
> We appreciate the reference to another current submission at ICLR, but given the inherent variability in pruning methods' performance across different sparsity levels, a comprehensive quantitative comparison based on sparsity-performance curves would be needed for a fair assessment.

---

### Official Review · Reviewer_i9Pr · 2023-10-30

**Soundness:** 4 excellent
**Presentation:** 3 good
**Contribution:** 2 fair
**Rating:** 5
**Confidence:** 4

**Summary:**

This paper proposed a method “Demon’s Pruning” (DemP) for structured pruning, which removes dead neurons during training. The paper studied the phenomenon of dying neurons during training and how the choices of hyperparameter configurations impact how dying neurons occur in neural networks. Experiment results on CIFAR-10 and ImageNet show the advantages of the proposed method DemP.

**Strengths:**

1. The paper studied the setting of dying neurons during training, which is missing in existing structured pruning approaches.
2. The perspective of learning a structured sparse network during training is interesting and promising.

**Weaknesses:**

1. No speedup evaluation is conducted. As a method for network pruning, it is expected to conduct real inference speedup evaluation over the original dense model.
2. It seems that the method is essentially a structured version of existing sparse training methods, e.g. RigL. It is not clear to me the technical contribution of this work.
3. The theory in section 3.2. seems not useful and redundant. It is not related to the method DemP itself.

**Questions:**

1. It is not clear to me why we should compare a structured pruning method with unstructured pruning methods, e.g. in Table 1. Since DemP is a structured pruning method, we should compare with possibly more structured pruning baselines?
2. Could the authors provide speedup evaluation of structured pruned models?
3. Could the authors illustrate clearly the difference of the “dynamic pruning” procedure in the paper and existing sparse training methods, e.g. RiGL and iterative magnitude pruning? It looks to me they are essentially the same except DemP is removing neurons while sparse training methods remove individual weights.
4. Adjust the theory part in section 3.2 and explain how it is related to DemP. It seems right now they are just some fancy equations which is not helpful for understanding the paper.

---

> ### Author Response · Authors · 2023-11-21
> **Response to Reviewer i9Pr**
>
> We thank the reviewer for their positive feedback on the presentation, soundness and novelty of our approach. More importantly, we thank the reviewer for their questions and concrete suggestions to clarify our contribution and improve the paper's presentation.
>
> ********************Questions:********************
>
> 1.  **Comparison with unstructured methods**.
>
> We appreciate the reviewer's valuable input regarding the comparison of DemP with unstructured pruning methods in Table 1. Our initial decision to include unstructured baselines was motivated by a desire for consistency with existing literature [1, 2], aiming to highlight the trade-off between structured and unstructured pruning approaches. However, we acknowledge the importance of contextualizing our comparisons within the domain of structured pruning.
>
> To address this concern, we have revised our manuscript in Section 5 to explicitly clarify the rationale behind including unstructured baselines. Additionally, in response to the reviewer's observation, we have relocated Figure 6 to Appendix I to maintain focus on the main results. Importantly, we want to emphasize that the structured baselines chosen for comparison represent, to the best of our knowledge, the top-performing dynamics pruning algorithms available.
>
> We refrained from introducing additional baselines that may offer subpar performance to prevent overcrowding the figures, ensuring a clear presentation of the most relevant and competitive comparisons. We believe these adjustments enhance the overall clarity and relevance of our comparisons, aligning more closely with the structured pruning context.
>
>  2.  **More speedup results**
>
> In line with the reviewer’s suggestion,  we added in Table 1 extended results for training speedup as well as training/inference FLOPs for the ResNet-50 architecture on ImageNet. We intend to run analogous experiments for the smaller networks as well and add the corresponding results to the camera-ready version.
>
> 3.  **Difference with existing methods**
>
> Thank you for this important question. Dynamic pruning is an overloaded expression, that we use in accordance to how it is used in RiGL paper. It consists of methods that gradually prune neural nets to recover a sparse model at the end of training under a fixed trained budget. It contrasts with methods doing single-shot pruning at the end of training before fine-tuning, or with iterative magnitude pruning that repeats multiple cycles of single-shot pruning after initial complete training. Apart from the structured nature of DemP, further differences are:
>
>  * DemP acts on the learning dynamics to force the optimization process to generate a sparse solution. It does not require single or multiple pruning interventions during training.
>   *  It alleviates the need to design a score function to identify the neurons to remove, monitoring the activations retrieved during the forward pass is sufficient.
>
>  We appreciate the opportunity to further highlight the originality of our approach. We made the necessary modifications in our paper’s introduction and in Section 4.
>
> 4. **On the relation between DemP and the theorerical analysis**
>
> We appreciate this important feedback.  Viewing the optimization process as akin to a biased random walk suggests that we can act on the learning dynamics to learn sparser solutions instead of relying on pruning intervention. DemP explores a single direction to affect the learning dynamics (regularization) while Section 3.2 was an attempt to further understand the impact of various hyperparameters on sparsity. We tried to clarify this connection by emphasizing this point of view in the introduction and section 4 and reworking the intuition. However we acknowledge that since the mathematical analysis is not central to the paper,  it tends to distracts away from the main point of the paper.  Following your suggestion, we recentered the discussion in section 3 on empirical evidence and moved the discussion of the Brownian motion model to Apppendix B, while  incorporating a reference to it in the intuition subsection of Section 3.1. We appreciate the opportunity to improve the flow and clarity of this analysis section.
>
> [1] John Rachwan, Daniel Zügner, Bertrand Charpentier, Simon Geisler, Morgane Ayle, Stephan Günnemann: **Winning the Lottery Ahead of Time: Efficient Early Network Pruning.** ICML 2022: 18293-18309
>
> [2] Stijn Verdenius, Maarten Stol, Patrick Forré: **Pruning via Iterative Ranking of Sensitivity Statistics.** CoRR abs/2006.00896 (2020)

---

### Official Review · Reviewer_k7gu · 2023-10-30

**Soundness:** 2 fair
**Presentation:** 2 fair
**Contribution:** 2 fair
**Rating:** 6
**Confidence:** 3

**Summary:**

This paper investigates the dying neurons phenomenon in network pruning. By employing the random walk model for network parameters, it reveals that neurons that become inactive during training may be challenging to recover. The study also explores how different hyperparameter configurations impact the occurrence of dying neurons. The authors thus introduce the "Demon’s Pruning" (DemP) to remove dead neurons in real time as they arise. This method dynamically prunes networks during training and outperforms some existing structured pruning techniques as shown in the experiments.

**Strengths:**

- Offers some insights into mechanisms of neuron mortality through the lens of network sparsity and pruning and provides analysis into the influence of gradient noise/learning rate/regularization. Experimental results support its findings to some extent.
- The proposed pruning method seems simple, computationally efficient, and straightforward to implement.

**Weaknesses:**

- The concept of pruning neural networks by eliminating inactive neurons based on their activation doesn't appear very novel. Several prior papers have explored the notion of dead neurons in sparse neural networks or proposed activation-based pruning methods [1,2,3].
- The analysis section seems to oversimplify the problem. It remains a question whether this analysis, built on Brownian motion model of weights rather than some model of activation, can effectively explain the activation-based pruning method.
- The paper is generally easy to follow, but certain sections could benefit from additional details to make it more self-contained and less confusing. For instance, providing background information on Brownian motion, discussing implicit assumptions when using absorbing Brownian motion model, and offering a clear definition of 'dead neurons' in the context of convolutional neural networks would enhance clarity.

[1] Hu, Hengyuan, et al. "Network trimming: A data-driven neuron pruning approach towards efficient deep architectures." *arXiv preprint arXiv:1607.03250* (2016).

[2] Whitaker, Tim, and Darrell Whitley. "Synaptic Stripping: How Pruning Can Bring Dead Neurons Back To Life." *arXiv preprint arXiv:2302.05818* (2023).

[3] Liu, Shiyu, Rohan Ghosh, and Mehul Motani. "AP: Selective Activation for De-sparsifying Pruned Networks." *Transactions on Machine Learning Research* (2023).

**Questions:**

- I'm curious about the definition of 'dead neurons' in convolutional neural networks and what specific structures DemP will remove.

- I'm also interested in understanding why a one-dimensional absorbing Brownian motion can effectively represent the weight dynamics of neural networks. Does the behavior of weights align with the assumptions of this model?

- The experiments employ a one-cycle scheduler for the regularization parameter. Is the comparison with other baseline methods also using the same regularization? It's important to consider that regularization may impact model performance.

---

> ### Author Response · Authors · 2023-11-21
> **Response to Reviewer k7gu**
>
> We thank the reviewer for the accurate summary of our work and for recognizing the simplicity and efficiency of the method we propose. We also appreciate the questions and constructive feedback to improve the flow of the paper.
>
> **Novelty of Our Approach:**
> We appreciate Reviewer K7GU's observation that the concepts of dead neurons and activation-based pruning methods have been explored in prior literature. However, we want to emphasize the distinctiveness of our contribution. To the best of our knowledge, our work is the first to demonstrate the viability and effectiveness of pruning based solely on units inactive across the entire dataset. This novel approach stems from our insights into the phenomenon, where we identified specific hyperparameters that make dead neurons prevalent enough for a successful pruning algorithm.
>
> Unlike previous methods [1,2,3], our approach uniquely ensures that the pruning action does not disrupt the learning dynamics, eliminating the need for recovery interventions. While [2] and [3] focused on restoring capacity through pruning negative weights, their emphasis aligns more closely with studies addressing plasticity loss and subsequent restoration [4,5].
>
> In reponse to the reviewer’s comment,  we further highlighted  these distinctive aspects in the revision,  in both the introduction and Section 4, providing a comprehensive context for the originality of our approach. We appreciate the opportunity to clarify these nuances.
>
> ********************Questions:********************
>
> 1. **Dead neurons definition for convolutional layers**. Thank you for pointing out this oversight.  We have added an explicit definition in Appendix A,  referenced in section 3 when we introduce the general definition. In convolutional layers, ReLU is applied element-wise to the pre-activation feature map. We consider an individual neuron (filter) dead if all elements of the feature map post-activation are 0.
> 2. **Brownian motion model**. The purpose of this model is to highlight in a simple setting the specific role played by the *noise structure* in the weights dynamics. Specifically,  the absorbing Brownian motion models a system subject to noise with ReLU activation structure;  and our analysis highlights how such a structured noise induced sparsity, as the weight ends up in the inactive region with probability 1.
>
>     We believe that this insight is important and reveals the pivotal role played by the stochasticity of the learning algorithm in the occurrence of dead neurons, as also illustrated in our empirical observations in Fig 2. However we acknowledge this is not central to the paper. As per Reviewer’s feedback,   (also per Reviewer i9Pr’s suggestion) we’ve moved the discussion of the Brownian motion model to Apppendix B and incorporated a reference to it in the intuition subsection of Section 3.1. We appreciate the opportunity to improve the flow and clarity of this analysis section.
>
>
> 1. **On the use of DemP’s regularization schedule for the baselines**. Thanks for highlighting this confusing aspect. Our baselines trained for comparison do not use the one-cycle regularization schedule used by DemP, but rather the constant weight decay suggested by the training recipes. We clarified this important aspect in Section 5 and in Appendix J1-J4.
>
> [1] Hu, Hengyuan, et al.: **Network trimming:** **A data-driven neuron pruning approach towards efficient deep architectures.** *arXiv preprint arXiv:1607.03250* (2016).
>
> [2] Whitaker, Tim, and Darrell Whitley: **Synaptic Stripping: How Pruning Can Bring Dead Neurons Back To Life.** *arXiv preprint arXiv:2302.05818* (2023).
>
> [3] Liu, Shiyu, Rohan Ghosh, and Mehul Motani: **AP: Selective Activation for De-sparsifying Pruned Networks.** *Transactions on Machine Learning Research* (2023).
>
> [4] Ghada Sokar, Rishabh Agarwal, Pablo Samuel Castro, Utku Evci: **The dormant neuron phenomenon in deep reinforcement learning.** ICML 2023: 32145-32168
>
> [5] Zaheer Abbas, Rosie Zhao, Joseph Modayil, Adam White, Marlos C. Machado: **Loss of Plasticity in Continual Deep Reinforcement Learning.** CoRR abs/2303.07507 (2023)

---

### Official Review · Reviewer_YsTs · 2023-11-01

**Soundness:** 3 good
**Presentation:** 3 good
**Contribution:** 2 fair
**Rating:** 6
**Confidence:** 3

**Summary:**

The manuscript provides mathematical and empirical support for the authors' argument that neuronal death is governed by settings of common hyperparameters (batch size, regularization strength, etc.). They leverage this observation by choosing hyperparameter settings that prompt neuron death, allowing them to prune such neurons to obtain training speedups. The results improve on those of other structured pruning algorithms on ResNet18, VGG16, and ResNet50.

**Strengths:**

Relative to other structured pruning algorithms, the proposed pruning approach (DemP) is more accurate and providing of stronger speedups.

The idea to prune dead neurons for efficient training is, as far as I know, original. If it can be shown to be helpful in more contexts (see "Questions" below), this method's simplicity and intuitive justification will make it impactful.

The manuscript provides a mathematical analysis in Section 3.2 that provides further intuition for the neuron death problem and its relationship to training hyperparameters, which is generally helpful.

The experiments (those analyzing neuron death and those analyzing DemP) are well designed and clear -- e.g., baseline structured pruning methods that are compared to DemP are thoughtfully chosen.

**Weaknesses:**

The experiments left unclear the practical relevance of DemP. As discussed more thoroughly below (see "Questions"), training in a wider variety of contexts and using more competitive training setups as baselines will help clarify whether DemP provides speedups that are relevant to readers' work/research.

**Questions:**

Score-affecting:

1. More results on speedups from DemP would be great to see. For instance, using your method, can you improve on the Mosaic ML ResNet50 ImageNet training time result? (See https://docs.mosaicml.com/projects/composer/en/stable/tutorials/train_resnet50_on_aws.html.)
2. ResNet18 typically gets ~95% accuracy on CIFAR10 but does not in your experiments. Could you please explain the hyperparameter choices causing the gap?
   - If the decrease in accuracy is caused by the regularization approach you use to encourage sparsity, that is seemingly a limitation of the proposed method that should be stated clearly.
3. In the main text, please make clear that the baseline methods compared to (e.g., EarlyCroP) are not using the (potentially suboptimal) regularization schedule required by DemP. I believe they are not based on my reading of Sections F.1 and F.3, but please correct me if I am wrong.
4. Is DemP effective on more modern architectures like ViTs or language models? Exploring transformer models need not require a significant increase in compute (e.g. results on GPT-2 Small would be interesting and that model is not too much larger than ResNet50).
5. The fact that unstructured pruning baselines aren't well tuned, which is noted in the manuscript, should coincide with more cautious claims about performance relative to unstructured pruning.
   - A well tuned magnitude pruning algorithm on ResNet-18 using CIFAR10 data actually improves baseline accuracy (95% accurate) at 95% sparsity (96% accurate); see Figure 1 of Diffenderfer et al. (2021).
   - Consider complementing Figure 6 with more details (in the main text) on the unstructured algorithms to provide needed context for their performances.
   - Relative to unstructured pruning, the benefit of DemP to emphasize is probably its ability to provide speedups (as opposed to its ability to match weakly-tuned unstructured pruning algorithms on accuracy).
6. It would be great to see ImageNet accuracy at a few different DemP sparsity levels in a figure that supplements Table 1 (i.e., show the accuracy-efficiency frontier created by DemP).
7. In Table 1, train your own baselines (at least for the dense model).
   - Right now, it's unclear what to attribute the gap between Dense and DemP to (training code differences or DemP's effect).


Important:
1. Figures should be closer to where they are discussed. E.g., Figure 4 is two pages away from where it's explained.

Minor:

1. The second paragraph of Section 3.1 should be made clearer -- where are the 3904 neurons that are referenced coming from?
2. The Maxwell's demon analogy in the intuition section is slightly unclear, perhaps clean this up a bit to avoid confusion.
   - If I understand correctly, the demon in that thought experiment is needed because there's otherwise a lack of a mechanism for entropy decreasing. When it comes to a neuron's transition to death/life, however, there are actual mechanisms (e.g., learning rate size) that govern transitions; no hypothetical demon is needed. Perhaps the function of the analogy is to clarify that, before the present manuscript, neuron death transition was treated too much like a black box (or demon). In any case, I suggest revising the "Intuition" paragraph that discusses this analogy.
3. At the top of page 5, a simplified version of Equation 4 is referenced, I think "3" is meant.
4. In equation 5, is eta in the denominator or numerator of the exponent? I think it's the latter but the typesetting leaves this unclear.
5. The last sentence of the third to last paragraph on page 5 is unclear, I think an "=0" is missing.
6. Figure 7 has an incorrect caption.
7. On page 8, you reference Table 5, I think you mean Table 1.

---

> ### Author Response · Authors · 2023-11-21
> **Response to Reviewer YsTs (1/2)**
>
> We thank the reviewer for their accurate summary of our work. We also thank the reviewer for insightful comments and concrete suggestions to improve the manuscript. We try to address as best as we can the issues raised below:
>
> 1.  **More speedup results**.  In line with the reviewer’s suggestion,  we added in Table 1 extended results for training speedup as well as training/inference FLOPs for the ResNet-50 architecture on ImageNet. We intend to run analogous experiments for the smaller networks as well and add the corresponding results to the camera-ready version.  Regarding the reviewer’s suggestion to refer to Mosaic ML’s results,  it is worth noting that these leverage a number of methods and tricks, such as  FFCV [1], in addition to pruning, so that fair,  direct comparisons are not straightforward. Moreover, the provided framework requires more computational resources than what we use in our experiments. Finally, it is only available in Pytorch where our code relies on JAX.
>
> 2. **Performances on CIFAR 10**. We appreciate the reviewer’s feedback, which highlights the need for clarifications in our manuscript.  First, it’s important to note that we report the test accuracy at the end of the training, while multiple works report the best accuracy reached during training (e.g., [3] sec. 5).  Second, while SGD+momentum typically gives better accuracies, we focused our experiments on ADAM because of its popularity and its inherent alignment with our method (see Appendix E). It is also the choice made by our strongest  structured pruning baseline [2].  The performance of our experiments seems aligned with the reported results in [3] (Fig. 5) for the ADAM optimizer.
>
> 3. **On the use of DemP’s regularization schedule for the baselines**. You are correct, our baselines do not use the (potentially suboptimal) regularization schedule used by DemP. We clarified this important aspect in Section 5 and in Appendix J1-J4.
>
> 4. **DemP on transformers**. While we acknowledge the importance of investigating the application of our approach to transformer-based models, we would like to highlight that our benchmarking was deliberately aligned with recent prior work on the topic (e.g., [2,4]). This alignment allows for a meaningful comparison with prior work, especially given our focus on gradual pruning during training to reduce computational costs.
>
> 5. **On our comparison with unstructured pruning baselines**. We thank the reviewer for these important comments and suggestions. We note that Diffenderfer et al. (2021) are targeting a different setup where pruning is done in a single shot after regular training, with additional fine-tuning to recover performance afterward. The goal we pursue with our method is to do dynamic pruning under a fixed training budget to reduce the computational load for learning the model. Also, they achieved this performance with SGD + momentum, while we used ADAM (see reply to question 2).
>
>  We agree with the reviewer that, relative to unstructured methods, the benefit of DemP is its ability to provide speedup. We included a quick comparison to unstructured methods that also prune the model throughout training to expose that the perceived trade-off between structured and unstructured methods (unstructured achieve better performance) is not always obvious and can require extensive fine-tuning. It also aligns with prior work [2].
>
>  Following your suggestion, we clarified further Fig. 6 with the points raised above. As per Reviewer i9Pr’c comment (question 1), we also moved this figure  (Figure 16 in the new revision) to Appendix I to focus the content on structured methods.
>
> 6. **More ImageNet results at different DemP sparsity levels**  Thank you for this great suggestion. We complemented Table 1 with further results, adding a comparison between methods at 90% sparsity as well. We also included the suggested figure in the revision (see Figure 7).
>
> 7.  **Implementation of Dense model**  Thanks again for pointing this inconsistency to our attention. Following your suggestion,  we have trained our own dense baseline and added the results to Table 1.
>
> **Please continue to the comment below**

---

> > ### Author Response · Authors · 2023-11-21
> > **Response to Reviewer YsTs (2/2)**
> >
> > **Position of Figures** Thank you for pointing out the misalignement of figures. We have reworked their position to better align them with where they are discussed.
> >
> > **Minor fixes**:
> >
> > 1. Thank you for this observation about the lack of explanation when evoking 3904 neurons. We clarified in the revision that in convolutional layers we consider the filters as individual neurons by appending a footnote to our definition in section 3 and further details in Appendix A. We also clarified further when evoking the 3904 neurons.
> >
> > 2. Thank you for bringing the issue of the Maxwell's demon analogy in the intuition section to our attention. We meant to personalize ReLU activations by a demon to connect with Maxwell’s thought experiment, which follows the same reasoning but in a thermodynamic context. As per the reviewer’s suggestion, we have mostly rewritten the intuition section to better emphasize the main ideas of the analogy.
> >
> >  3-7: Thanks for catching those errors and ambiguities.  We have corrected them accordingly.
> >
> > [1] Guillaume Leclerc, Andrew Ilyas, Logan Engstrom, Sung Min Park, Hadi Salman, Aleksander Madry: **FFCV: Accelerating Training by Removing Data Bottlenecks.** CVPR 2023: 12011-12020
> >
> > [2] John Rachwan, Daniel Zügner, Bertrand Charpentier, Simon Geisler, Morgane Ayle, Stephan Günnemann: **Winning the Lottery Ahead of Time: Efficient Early Network Pruning.** ICML 2022: 18293-18309
> >
> > [3] Thomas Moreau et al.: **Benchopt: Reproducible, efficient and collaborative optimization benchmarks.** NeurIPS 2022
> >
> > [4] Utku Evci, Trevor Gale, Jacob Menick, Pablo Samuel Castro, Erich Elsen: **Rigging the Lottery: Making All Tickets Winners.** ICML 2020: 2943-2952

---

> ### Comment · Reviewer_YsTs · 2023-11-23
>
> Thank you for adding the new results and clarifications! The experiments appear more convincing now, and the speedups are particularly nice to see.
>
> In main text Figures like 4b, 5b, 6b, and 7b (especially Figure 6b), I think a plot of training speedup vs. "neuron sparsity" would be more compelling than plotting Accuracy vs. "weight sparsity" (inference speedup results would be great too). For example, if you show a scenario where accuracy is not significantly harmed (<1% difference) and training time is improved (e.g., by 5+%), that would increase the relevance of this paper to the efficient training literature (e.g., see "Compute-Efficient Deep Learning", Bartoldson et al., 2023). Relatedly, extending the experiments to show speedups on transformer training could greatly raise the contribution of this manuscript.
>
> I have raised my score. Going forward, I believe this paper can attain a significantly broader impact by expanding its focus in the suggested ways (improving on the MosaicML or another benchmark ResNet50 time, showing applications to transformer training, etc.).

---

### Author Response · Authors · 2023-11-21
**General Response**

We are grateful to all reviewers for their engaged, insightful and concrete feedback, which helped us greatly improve our manuscript. We are encouraged by the reviewers' recognition of the soudness of our approach (e.g. Reviewer i9Pr) and its originality (Reviewer YsTs),  the quality of our experimental design and choice of baselines (Reviewer YsT and tiWA) and the simplicity and efficiency of our pruning method (Reviewers k7gu and tiWA).

In response to the reviewers' feedback, we have worked to address most of their concerns and incorporated their suggestions in the revised version. Notable enhancements include additional experiments to assess speedup benefits, including FLOPs calculation (refer to Table 1), and extended results for ResNet-50 pruning across various sparsity levels (refer to Figure 6). To better clarify the distinctions of our method from existing approaches, we have provided detailed explanations in both the introduction and Section 5. The analysis section has been reworked for improved clarity, and we have carefully revised potentially misleading sentences throughout the manuscript.

For the convenience of reviewers, changes in the updated manuscript are highlighted in red in the revised PDF. We firmly believe that these modifications significantly strengthen the contribution of our submission and enhance the clarity of its presentation. We sincerely appreciate the thorough review process, which has undoubtedly contributed to the overall refinement of our work.

---

### Meta-Review · Area_Chair_1uwA · 2023-12-14

**Metareview:**

This research explores the issue of dying neurons in the context of network pruning. Utilizing a random walk model for network parameters, the paper uncovers that neurons turning inactive during training could be difficult to reactivate. It also examines the influence of various hyperparameter settings on the prevalence of dying neurons. Consequently, the authors propose "Demon’s Pruning" (DemP), a technique to eliminate dead neurons in real-time as they emerge.

Although most reviewers find that the exploration of dying neurons in training neural networks insightful and promising, there are several concerns on the current results need to be further addressed, including the motivation, the oversimplification of theoretical study, and a better discussion and position compared to existing work on sparse training and pruning. We recommend the authors resubmit to other conferences by incorporating all reviewers’ feedbacks.

**Justification For Why Not Higher Score:**

The current version of the paper needs improvement before publication.

**Justification For Why Not Lower Score:**

N/A

---

### Decision · Program_Chairs · 2024-01-16

Reject